# Quantification of Uncertainty with Adversarial Models

**Kajetan Schweighofer**[*]  **Lukas Aichberger**[*]  **Mykyta Ielanskyi**[*]
**Günter Klambauer**   **Sepp Hochreiter**

ELLIS Unit Linz and LIT AI Lab, Institute for Machine Learning,
Johannes Kepler University Linz, Austria
[*]Joint first authors

## Abstract

Quantifying uncertainty is important for actionable predictions in real-world applications. A crucial part of predictive uncertainty quantification is the estimation of epistemic uncertainty, which is defined as an integral of the product between a divergence function and the posterior. Current methods such as Deep Ensembles or MC dropout underperform at estimating the epistemic uncertainty, since they primarily consider the posterior when sampling models. We suggest Quantification of Uncertainty with Adversarial Models (QUAM) to better estimate the epistemic uncertainty. QUAM identifies regions where the whole product under the integral is large, not just the posterior. Consequently, QUAM has lower approximation error of the epistemic uncertainty compared to previous methods. Models for which the product is large correspond to adversarial models (not adversarial examples!). Adversarial models have both a high posterior as well as a high divergence between their predictions and that of a reference model. Our experiments show that QUAM excels in capturing epistemic uncertainty for deep learning models and outperforms previous methods on challenging tasks in the vision domain.

## 1   Introduction

Actionable predictions typically require risk assessment based on predictive uncertainty quantification [Apostolakis, 1991]. This is of utmost importance in high stake applications, such as medical diagnosis or drug discovery, where human lives or extensive investments are at risk. In such settings, even a single prediction has far-reaching real-world impact, thus necessitating the most precise quantification of the associated uncertainties. Furthermore, foundation models or specialized models that are obtained externally are becoming increasingly prevalent, also in high stake applications. It is crucial to assess the robustness and reliability of those unknown models before applying them. Therefore, the predictive uncertainty of given, pre-selected models at specific test points should be quantified, which we address in this work.

We consider predictive uncertainty quantification (see Fig. 1) for deep neural networks [Gal, 2016, Hüllermeier and Waegeman, 2021]. According to Vesely and Rasmuson [1984], Apostolakis [1991], Helton [1993], McKone [1994], Helton [1997], predictive uncertainty can be categorized into two types. First, *aleatoric* (Type A, variability, stochastic, true, irreducible) uncertainty refers to the variability when drawing samples or when repeating the same experiment. Second, *epistemic* (Type B, lack of knowledge, subjective, reducible) uncertainty refers to the lack of knowledge about the true model. Epistemic uncertainty can result from imprecision in parameter estimates, incompleteness in modeling, or indefiniteness in the applicability of the model. While aleatoric uncertainty cannot be reduced, epistemic uncertainty can be reduced by more data, better models, or more knowledge about the problem. We follow Helton [1997] and consider epistemic uncertainty as the imprecision or variability of parameters that determine the predictive distribution. Vesely and Rasmuson [1984]

37th Conference on Neural Information Processing Systems (NeurIPS 2023).

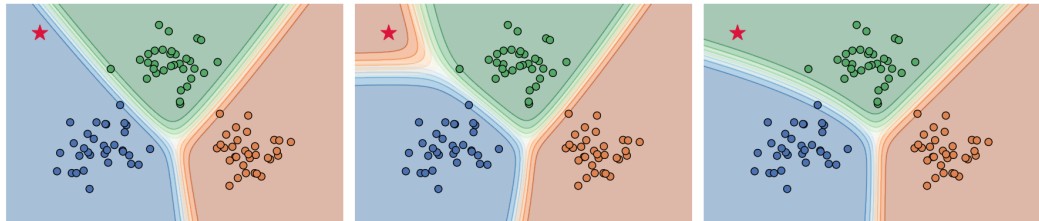

Figure 1: Adversarial models. For the red test point, the predictive uncertainty is high as it is far from the training data. High uncertainties are detected by different adversarial models that assign the red test point to different classes, although all of them explain the training data equally well. As a result, the true class of the test point remains ambiguous.

calls this epistemic uncertainty "parameter uncertainty", which results from an imperfect learning algorithm or from insufficiently many training samples. Consequently, we consider predictive uncertainty quantification as characterizing a probabilistic model of the world. In this context, aleatoric uncertainty refers to the inherent stochasticity of sampling outcomes from the predictive distribution of the model and epistemic uncertainty refers to the uncertainty about model parameters.

Current uncertainty quantification methods such as Deep Ensembles [Lakshminarayanan et al., 2017] or Monte-Carlo (MC) dropout [Gal and Ghahramani, 2016] underperform at estimating the epistemic uncertainty [Wilson and Izmailov, 2020, Parker-Holder et al., 2020, Angelo and Fortuin, 2021], since they primarily consider the posterior when sampling models. Thus they are prone to miss important posterior modes, where the whole integrand of the integral defining the epistemic uncertainty is large. We introduce Quantification of Uncertainty with Adversarial Models (QUAM) to identify those posterior modes. QUAM searches for those posterior modes via adversarial models and uses them to reduce the approximation error when estimating the integral that defines the epistemic uncertainty.

Adversarial models are characterized by a large value of the integrand of the integral defining the epistemic uncertainty. Thus, they considerably differ to the reference model's prediction at a test point while having a similarly high posterior probability. Consequently, they are counterexamples of the reference model that predict differently for a new input, but explain the training data equally well. Fig. 1 shows examples of adversarial models which assign different classes to a test point, but agree on the training data. A formal definition is given by Def. 1. It is essential to note that adversarial models are a new concept that is to be distinguished from other concepts that include the term 'adversarial' in their naming, such as adversarial examples [Szegedy et al., 2013, Biggio et al., 2013], adversarial training [Goodfellow et al., 2015], generative adversarial networks [Goodfellow et al., 2014] or adversarial model-based RL [Rigter et al., 2022].

Our main contributions are:

- We introduce QUAM as a framework for uncertainty quantification. QUAM approximates the integral that defines the epistemic uncertainty substantially better than previous methods, since it reduces the approximation error of the integral estimator.
- We introduce the concept of adversarial models for estimating posterior integrals with non-negative integrands. For a given test point, adversarial models have considerably different predictions than a reference model while having similarly high posterior probability.
- We introduce a new setting for uncertainty quantification, where the uncertainty of a given, pre-selected model is quantified.

## 2 Current Methods to Estimate the Epistemic Uncertainty

**Definition of Predictive Uncertainty.** Predictive uncertainty quantification is about describing a probabilistic model of the world, where aleatoric uncertainty refers to the inherent stochasticity of sampling outcomes from the predictive distribution of the model and epistemic uncertainty refers to the uncertainty about model parameters. We consider two distinct settings of predictive uncertainty quantification. Setting **(a)** concerns with the predictive uncertainty at a new test point expected under all plausible models given the training dataset [Gal, 2016, Hüllermeier and Waegeman, 2021]. This definition of uncertainty comprises how differently possible models predict (epistemic) and how

confident each model is about its prediction (aleatoric). Setting **(b)** concerns with the predictive uncertainty at a new test point for a given, pre-selected model. This definition of uncertainty comprises how likely this model is the true model that generated the training dataset (epistemic) [Apostolakis, 1991, Helton, 1997] and how confident this model is about its prediction (aleatoric).

As an example, assume we have initial data from an epidemic, but we do not know the exact infection rate, which is a parameter of a prediction model. The goal is to predict the number of infected persons at a specific time in the future, where each point in time is a test point. In setting (a), we are interested in the uncertainty of test point predictions of all models using infection rates that explain the initial data. If all likely models agree for a given new test point, the prediction of any of those models can be trusted, otherwise we can not trust the prediction regardless of which model is selected in the end. In setting (b), we have selected a specific infection rate from the initial data as parameter for our model to make predictions. We refer to this model as the given, pre-selected model. However, we do not know the true infection rate of the epidemic. All models with infection rates that are consistent with the initial data are likely to be the true model. If all likely models agree with the given, pre-selected model for a given new test point, the prediction of the model can be trusted.

## 2.1 Measuring Predictive Uncertainty

We consider the predictive distribution of a single model $p(\boldsymbol{y} \mid \boldsymbol{x}, \boldsymbol{w})$, which is a probabilistic model of the world. Depending on the task, the predictive distribution of this probabilistic model can be a categorical distribution for classification or a Gaussian distribution for regression. The Bayesian framework offers a principled way to treat the uncertainty about the parameters through the posterior $p(\boldsymbol{w} \mid \mathcal{D}) \propto p(\mathcal{D} \mid \boldsymbol{w})p(\boldsymbol{w})$ for a given dataset $\mathcal{D}$. The Bayesian model average (BMA) predictive distribution is given by $p(\boldsymbol{y} \mid \boldsymbol{x}, \mathcal{D}) = \int_{\mathcal{W}} p(\boldsymbol{y} \mid \boldsymbol{x}, \tilde{\boldsymbol{w}})p(\tilde{\boldsymbol{w}} \mid \mathcal{D})\mathrm{d}\tilde{\boldsymbol{w}}$. Following Gal [2016], Depeweg et al. [2018], Smith and Gal [2018], Hüllermeier and Waegeman [2021], the uncertainty of the BMA predictive distribution is commonly measured by the entropy $\mathrm{H}[p(\boldsymbol{y} \mid \boldsymbol{x}, \mathcal{D})]$. It refers to the total uncertainty, which can be decomposed into an aleatoric and an epistemic part. The BMA predictive entropy is equal to the posterior expectation of the cross-entropy $\mathrm{CE}[\cdot, \cdot]$ between the predictive distribution of candidate models and the BMA, which corresponds to setting (a). In setting (b), the cross-entropy is between the predictive distribution of the given, pre-selected model and candidate models. Details about the entropy and cross-entropy as measures of uncertainty are given in Sec. B.1.1 in the appendix. In the following, we formalize how to measure the notions of uncertainty in setting (a) and (b) using the expected cross-entropy over the posterior.

**Setting (a): Expected uncertainty when selecting a model.** We estimate the predictive uncertainty at a test point $\boldsymbol{x}$ when selecting a model $\tilde{\boldsymbol{w}}$ given a training dataset $\mathcal{D}$. The total uncertainty is the expected cross-entropy between the predictive distribution of candidate models $p(\boldsymbol{y} \mid \boldsymbol{x}, \tilde{\boldsymbol{w}})$ and the BMA predictive distribution $p(\boldsymbol{y} \mid \boldsymbol{x}, \mathcal{D})$, where the expectation is with respect to the posterior:

$$\int_{\mathcal{W}} \mathrm{CE}[p(\boldsymbol{y} \mid \boldsymbol{x}, \tilde{\boldsymbol{w}}), \, p(\boldsymbol{y} \mid \boldsymbol{x}, \mathcal{D})] \, p(\tilde{\boldsymbol{w}} \mid \mathcal{D}) \, \mathrm{d}\tilde{\boldsymbol{w}} \; = \; \mathrm{H}[p(\boldsymbol{y} \mid \boldsymbol{x}, \mathcal{D})] \tag{1}$$

$$= \int_{\mathcal{W}} \mathrm{H}[p(\boldsymbol{y} \mid \boldsymbol{x}, \tilde{\boldsymbol{w}})] \, p(\tilde{\boldsymbol{w}} \mid \mathcal{D}) \, \mathrm{d}\tilde{\boldsymbol{w}} \; + \; \mathrm{I}[Y \, ; \, W \mid \boldsymbol{x}, \mathcal{D}]$$

$$= \underbrace{\int_{\mathcal{W}} \mathrm{H}[p(\boldsymbol{y} \mid \boldsymbol{x}, \tilde{\boldsymbol{w}})] \, p(\tilde{\boldsymbol{w}} \mid \mathcal{D}) \, \mathrm{d}\tilde{\boldsymbol{w}}}_{\text{aleatoric}} \; + \; \underbrace{\int_{\mathcal{W}} \mathrm{D}_{\mathrm{KL}}(p(\boldsymbol{y} \mid \boldsymbol{x}, \tilde{\boldsymbol{w}}) \, \| \, p(\boldsymbol{y} \mid \boldsymbol{x}, \mathcal{D})) \, p(\tilde{\boldsymbol{w}} \mid \mathcal{D}) \, \mathrm{d}\tilde{\boldsymbol{w}}}_{\text{epistemic}} \; .$$

The aleatoric uncertainty characterizes the uncertainty due to the expected stochasticity of sampling outcomes from the predictive distribution of candidate models $p(\boldsymbol{y} \mid \boldsymbol{x}, \tilde{\boldsymbol{w}})$. The epistemic uncertainty characterizes the uncertainty due to the mismatch between the predictive distribution of candidate models and the BMA predictive distribution. It is measured by the mutual information $\mathrm{I}[\cdot \, ; \, \cdot]$, between the prediction $Y$ and the model parameters $W$ for a given test point and dataset, which is equivalent to the posterior expectation of the KL-divergence $\mathrm{D}_{\mathrm{KL}}(\cdot \, \| \, \cdot)$ between the predictive distributions of candidate models and the BMA predictive distribution. Derivations are given in appendix Sec. B.1.

**Setting (b): Uncertainty of a given, pre-selected model.** We estimate the predictive uncertainty of a given, pre-selected model $\boldsymbol{w}$ at a test point $\boldsymbol{x}$. We assume that the dataset $\mathcal{D}$ is produced according to the true distribution $p(\boldsymbol{y} \mid \boldsymbol{x}, \boldsymbol{w}^*)$ parameterized by $\boldsymbol{w}^*$. The posterior $p(\tilde{\boldsymbol{w}} \mid \mathcal{D})$ is an estimate of

how likely $\tilde{w}$ match $w^*$. For epistemic uncertainty, we should measure the difference between the predictive distributions under $w$ and $w^*$, but $w^*$ is unknown. Therefore, we measure the expected difference between the predictive distributions under $w$ and $\tilde{w}$. In accordance with Apostolakis [1991] and Helton [1997], the total uncertainty is therefore the expected cross-entropy between the predictive distributions of a given, pre-selected model $w$ and candidate models $\tilde{w}$, any of which could be the true model $w^*$ according to the posterior:

$$\int_{\mathcal{W}} \mathrm{CE}[p(\boldsymbol{y} \mid \boldsymbol{x}, \boldsymbol{w}),\, p(\boldsymbol{y} \mid \boldsymbol{x}, \tilde{\boldsymbol{w}})]\, p(\tilde{\boldsymbol{w}} \mid \mathcal{D})\, \mathrm{d}\tilde{\boldsymbol{w}} \tag{2}$$

$$= \underbrace{\mathrm{H}[p(\boldsymbol{y} \mid \boldsymbol{x}, \boldsymbol{w})]}_{\text{aleatoric}} + \underbrace{\int_{\mathcal{W}} \mathrm{D}_{\mathrm{KL}}(p(\boldsymbol{y} \mid \boldsymbol{x}, \boldsymbol{w}) \,\|\, p(\boldsymbol{y} \mid \boldsymbol{x}, \tilde{\boldsymbol{w}}))\, p(\tilde{\boldsymbol{w}} \mid \mathcal{D})\, \mathrm{d}\tilde{\boldsymbol{w}}}_{\text{epistemic}}\ .$$

The aleatoric uncertainty characterizes the uncertainty due to the stochasticity of sampling outcomes from the predictive distribution of the given, pre-selected model $p(\boldsymbol{y} \mid \boldsymbol{x}, \boldsymbol{w})$. The epistemic uncertainty characterizes the uncertainty due to the mismatch between the predictive distribution of the given, pre-selected model and the predictive distribution of candidate models that could be the true model. Derivations and further details are given in appendix Sec. B.1.

## 2.2 Estimating the Integral for Epistemic Uncertainty

Current methods for predictive uncertainty quantification suffer from underestimating the epistemic uncertainty [Wilson and Izmailov, 2020, Parker-Holder et al., 2020, Angelo and Fortuin, 2021]. The epistemic uncertainty is given by the respective terms in Eq. (1) for setting (a) and Eq. (2) for our new setting (b). To estimate these integrals, almost all methods use gradient descent on the training data. Thus, posterior modes that are hidden from the gradient flow remain undiscovered and the epistemic uncertainty is underestimated [Shah et al., 2020, Angelo and Fortuin, 2021]. An illustrative example is depicted in Fig. 2. Posterior expectations as in Eq. (1) and Eq. (2) that define the epistemic uncertainty are generally approximated using Monte Carlo integration. A good approximation of posterior integrals through Monte Carlo integration requires to capture all large values of the non-negative integrand [Wilson and Izmailov, 2020], which is not only large values of the posterior, but also large values of the KL-divergence.

Variational inference [Graves, 2011, Blundell et al., 2015, Gal and Ghahramani, 2016] and ensemble methods [Lakshminarayanan et al., 2017] estimate the posterior integral based on models with high posterior. Posterior modes may be hidden from gradient descent based techniques as they only discover mechanistically similar models. Two models are mechanistically similar if they rely on the same input attributes for making their predictions, that is, they are invariant to the same input attributes [Lubana et al., 2022]. However, gradient descent will always start by extracting input attributes that are highly correlated to the target as they determine the steepest descent in the error

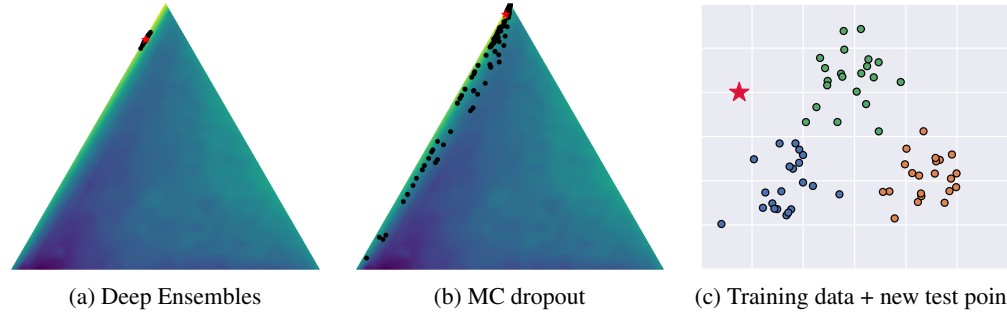

(a) Deep Ensembles      (b) MC dropout      (c) Training data + new test point

Figure 2: Model prediction analysis. Softmax outputs (black) of individual models of Deep Ensembles (a) and MC dropout (b), as well as their average output (red) on a probability simplex. Models were selected on the training data, and evaluated on the new test point (red) depicted in (c). The background color denotes the maximum likelihood of the training data that is achievable by a model having a predictive distribution (softmax values) equal to the respective location on the simplex. Deep Ensembles and MC dropout fail to find models predicting the orange class, although there would be likely models that do so. Details on the experimental setup are given in the appendix, Sec. C.2.

landscape. These input attributes create a large basin in the error landscape into which the parameter vector is drawn via gradient descent. Consequently, other modes further away from such basins are almost never found [Shah et al., 2020, Angelo and Fortuin, 2021]. Thus, the epistemic uncertainty is underestimated. Another reason that posterior modes may be hidden from gradient descent is the presence of different labeling hypotheses. If there is more than one way to explain the training data, gradient descent will use all of them as they give the steepest error descent [Scimeca et al., 2022].

Other work focuses on MCMC sampling according to the posterior distribution, which is approximated by stochastic gradient variants [Welling and Teh, 2011, Chen et al., 2014] for large datasets and models. Those are known to face issues to efficiently explore the highly complex and multimodal parameter space and escape local posterior modes. There are attempts to alleviate the problem [Li et al., 2016, Zhang et al., 2020]. However, those methods do not explicitly look for important posterior modes, where the predictive distributions of sampled models contribute strongly to the approximation of the posterior integral, and thus have large values for the KL-divergence.

## 3 Adversarial Models to Estimate the Epistemic Uncertainty

**Intuition.** The epistemic uncertainty in Eq. (1) for setting (a) compares possible models with the BMA. Thus, the BMA is used as reference model. The epistemic uncertainty in Eq. (2) for our new setting (b) compares models that are candidates for the true model with the given, pre-selected model. Thus, the given, pre-selected model is used as reference model. If the reference model makes some prediction at the test point, and if other models (the adversaries) make different predictions while explaining the training data equally well, then one should be uncertain about the prediction. Adversarial models are plausible outcomes of model selection, while having a different prediction at the test data point than the reference model. In court, the same principle is used: if the prosecutor presents a scenario but the advocate presents alternative equally plausible scenarios, the judges become uncertain about what happened and rule in favor of the defendant. We use adversarial models to identify locations where the integrand of the integral defining the epistemic uncertainty in Eq. (1) or Eq. (2) is large. These locations are used to construct a mixture distribution that is used for mixture importance sampling to estimate the desired integrals. Using the mixture distribution for sampling, we aim to considerably reduce the approximation error of the estimator of the epistemic uncertainty.

**Mixture Importance Sampling.** We estimate the integrals of epistemic uncertainty in Eq. (1) and in Eq. (2). In the following, we focus on setting (b) with Eq. (2), but all results hold for setting (a) with Eq. (1) as well. Most methods sample from a distribution $q(\tilde{\boldsymbol{w}})$ to approximate the integral:

$$v = \int_{\mathcal{W}} \mathrm{D}_{\mathrm{KL}}(p(\boldsymbol{y} \mid \boldsymbol{x}, \boldsymbol{w}) \,\|\, p(\boldsymbol{y} \mid \boldsymbol{x}, \tilde{\boldsymbol{w}})) \, p(\tilde{\boldsymbol{w}} \mid \mathcal{D}) \, \mathrm{d}\tilde{\boldsymbol{w}} = \int_{\mathcal{W}} \frac{u(\boldsymbol{x}, \boldsymbol{w}, \tilde{\boldsymbol{w}})}{q(\tilde{\boldsymbol{w}})} \, q(\tilde{\boldsymbol{w}}) \, \mathrm{d}\tilde{\boldsymbol{w}} \,, \quad (3)$$

where $u(\boldsymbol{x}, \boldsymbol{w}, \tilde{\boldsymbol{w}}) = \mathrm{D}_{\mathrm{KL}}(p(\boldsymbol{y} \mid \boldsymbol{x}, \boldsymbol{w}) \,\|\, p(\boldsymbol{y} \mid \boldsymbol{x}, \tilde{\boldsymbol{w}})) p(\tilde{\boldsymbol{w}} \mid \mathcal{D})$. As with Deep Ensembles or MC dropout, posterior sampling is often approximated by a sampling distribution $q(\tilde{\boldsymbol{w}})$ that is close to $p(\tilde{\boldsymbol{w}} \mid \mathcal{D})$. Monte Carlo (MC) integration estimates $v$ by

$$\hat{v} = \frac{1}{N} \sum_{n=1}^{N} \frac{u(\boldsymbol{x}, \boldsymbol{w}, \tilde{\boldsymbol{w}}_n)}{q(\tilde{\boldsymbol{w}}_n)} \,, \qquad \tilde{\boldsymbol{w}}_n \sim q(\tilde{\boldsymbol{w}}) \,. \quad (4)$$

If the posterior has different modes, the estimate under a unimodal approximate distribution has high variance and converges very slowly [Steele et al., 2006]. Thus, we use mixture importance sampling (MIS) [Hesterberg, 1995]. MIS utilizes a mixture distribution instead of the unimodal distribution in standard importance sampling [Owen and Zhou, 2000]. Furthermore, many MIS methods iteratively enhance the sampling distribution by incorporating new modes [Raftery and Bao, 2010]. In contrast to the usually applied iterative enrichment methods which find new modes by chance, we have a much more favorable situation. We can explicitly search for posterior modes where the KL divergence is large, as we can cast it as a supervised learning problem. Each of these modes determines the location of a mixture component of the mixture distribution.

**Theorem 1.** *The expected mean squared error of importance sampling with $q(\tilde{\boldsymbol{w}})$ can be bounded by*

$$\mathrm{E}_{q(\tilde{\boldsymbol{w}})} \left[ (\hat{v} - v)^2 \right] \leqslant \mathrm{E}_{q(\tilde{\boldsymbol{w}})} \left[ \left( \frac{u(\boldsymbol{x}, \boldsymbol{w}, \tilde{\boldsymbol{w}})}{q(\tilde{\boldsymbol{w}})} \right)^2 \right] \frac{4}{N} \,. \quad (5)$$

*Proof.* The inequality Eq. (5) follows from Theorem 1 in Akyildiz and Míguez [2021], when considering $0 \leqslant u(\boldsymbol{x}, \boldsymbol{w}, \tilde{\boldsymbol{w}})$ as an unnormalized distribution and setting $\varphi = 1$. $\qquad\square$

Approximating only the posterior $p(\tilde{\boldsymbol{w}} \mid \mathcal{D})$ as done by Deep Ensembles or MC dropout is insufficient to guarantee a low expected mean squared error, since the sampling variance cannot be bounded (see appendix Sec. B.2).

**Corollary 1.** *With constant $c$, $\mathrm{E}_{q(\tilde{\boldsymbol{w}})}\left[(\hat{v} - v)^2\right] \leqslant 4c^2/N$ holds if $u(\boldsymbol{x}, \boldsymbol{w}, \tilde{\boldsymbol{w}}) \leqslant c\, q(\tilde{\boldsymbol{w}})$.*

Consequently, $q(\tilde{\boldsymbol{w}})$ must have modes where $u(\boldsymbol{x}, \boldsymbol{w}, \tilde{\boldsymbol{w}})$ has modes even if the $q$-modes are a factor $c$ smaller. The modes of $u(\boldsymbol{x}, \boldsymbol{w}, \tilde{\boldsymbol{w}})$ are models $\tilde{\boldsymbol{w}}$ with both high posterior and high KL-divergence. We are searching for these modes to determine the locations $\breve{\boldsymbol{w}}_k$ of the components of a mixture distribution $q(\tilde{\boldsymbol{w}})$:

$$q(\tilde{\boldsymbol{w}}) = \sum_{k=1}^{K} \alpha_k\, \mathcal{P}(\tilde{\boldsymbol{w}}\, ; \breve{\boldsymbol{w}}_k, \boldsymbol{\theta}) , \qquad (6)$$

with $\alpha_k = 1/K$ for $K$ such models $\breve{\boldsymbol{w}}_k$ that determine a mode. Adversarial model search finds the locations $\breve{\boldsymbol{w}}_k$ of the mixture components, where $\breve{\boldsymbol{w}}_k$ is an adversarial model. The reference model does not define a mixture component, as it has zero KL-divergence to itself. We then sample from a distribution $\mathcal{P}$ at the local posterior mode with mean $\breve{\boldsymbol{w}}_k$ and a set of shape parameters $\boldsymbol{\theta}$. The simplest choice for $\mathcal{P}$ is a Dirac delta distribution, but one could use e.g. a local Laplace approximation of the posterior [MacKay, 1992], or a Gaussian distribution in some weight-subspace [Maddox et al., 2019]. Furthermore, one could use $\breve{\boldsymbol{w}}_k$ as starting point for SG-MCMC chains [Welling and Teh, 2011, Chen et al., 2014, Zhang et al., 2020, 2022]. More details regarding MIS are given in the appendix in Sec. B.2. In the following, we propose an algorithm to find those models with both high posterior and high KL-divergence to the predictive distribution of the reference model.

**Adversarial Model Search.** Adversarial model search is the concept of searching for a model that has a large distance / divergence to the reference predictive distribution and at the same time a high posterior. We call such models "adversarial models" as they act as adversaries to the reference model by contradicting its prediction. A formal definition of an adversarial model is given by Def. 1:

**Definition 1.** *Given are a new test data point $\boldsymbol{x}$, a reference conditional probability model $p(\boldsymbol{y} \mid \boldsymbol{x}, \boldsymbol{w})$ from a model class parameterized by $\boldsymbol{w}$, a divergence or distance measure $\mathrm{D}(\cdot, \cdot)$ for probability distributions, $\gamma > 0$, $\Lambda > 0$, and a dataset $\mathcal{D}$. Then a model with parameters $\tilde{\boldsymbol{w}}$ that satisfies the inequalities $|\log p(\boldsymbol{w} \mid \mathcal{D}) - \log p(\tilde{\boldsymbol{w}} \mid \mathcal{D})| \leqslant \gamma$ and $\mathrm{D}(p(\boldsymbol{y} \mid \boldsymbol{x}, \boldsymbol{w}), p(\boldsymbol{y} \mid \boldsymbol{x}, \tilde{\boldsymbol{w}})) \geq \Lambda$ is called an $(\gamma, \Lambda)-$adversarial model.*

Adversarial model search corresponds to the following optimization problem:

$$\max_{\boldsymbol{\delta} \in \Delta} \mathrm{D}(p(\boldsymbol{y} \mid \boldsymbol{x}, \boldsymbol{w}), p(\boldsymbol{y} \mid \boldsymbol{x}, \boldsymbol{w} + \boldsymbol{\delta})) \quad \text{s.t. } \log p(\boldsymbol{w} \mid \mathcal{D}) - \log p(\boldsymbol{w} + \boldsymbol{\delta} \mid \mathcal{D}) \leqslant \gamma . \quad (7)$$

We are searching for a weight perturbation $\boldsymbol{\delta}$ that maximizes the distance $\mathrm{D}(\cdot, \cdot)$ to the reference distribution without decreasing the log posterior more than $\gamma$. The search for adversarial models is restricted to $\boldsymbol{\delta} \in \Delta$, for example by only optimizing the last layer of the reference model or by bounding the norm of $\boldsymbol{\delta}$. This optimization problem can be rewritten as:

$$\max_{\boldsymbol{\delta} \in \Delta} \mathrm{D}(p(\boldsymbol{y} \mid \boldsymbol{x}, \boldsymbol{w}), p(\boldsymbol{y} \mid \boldsymbol{x}, \boldsymbol{w} + \boldsymbol{\delta})) + c\, (\log p(\boldsymbol{w} + \boldsymbol{\delta} \mid \mathcal{D}) - \log p(\boldsymbol{w} \mid \mathcal{D}) + \gamma) . \quad (8)$$

where $c$ is a hyperparameter. According to the *Karush-Kuhn-Tucker (KKT) theorem* [Karush, 1939, Kuhn and Tucker, 1950, May, 2020, Luenberger and Ye, 2016]: If $\boldsymbol{\delta}^*$ is the solution to the problem Eq. (7), then there exists a $c^* \geq 0$ with $\nabla_{\boldsymbol{\delta}} \mathcal{L}(\boldsymbol{\delta}^*, c^*) = \boldsymbol{0}$ ($\mathcal{L}$ is the Lagrangian) and $c^*\, (\log p(\boldsymbol{w} \mid \mathcal{D}) - \log p(\boldsymbol{w} + \boldsymbol{\delta}^* \mid \mathcal{D}) - \gamma) = 0$. This is a necessary condition for an optimal point according to Theorem on Page 326 of Luenberger and Ye [2016].

We solve this optimization problem by the penalty method, which relies on the KKT theorem [Zangwill, 1967]. A penalty algorithm solves a series of unconstrained problems, solutions of which converge to the solution of the original constrained problem (see e.g. Fiacco and McCormick [1990]). The unconstrained problems are constructed by adding a weighted penalty function measuring the constraint violation to the objective function. At every step, the weight of the penalty is increased, thus the constraints are less violated. If exists, the solution to the constraint optimization problem is an adversarial model that is located within a posterior mode but has a different predictive distribution compared to the reference model. We summarize the adversarial model search in Alg. 1.

---

**Algorithm 1** Adversarial Model Search (used in QUAM)

---

**Supplies:** Adversarial model $\breve{w}$ with maximum $L_{adv}$ and $L_{pen} \leqslant 0$

**Requires:** Test point $x$, training dataset $\mathcal{D} = \{(x_k, y_k)\}_{k=1}^K$, reference model $w$, loss function $l$, loss of reference model on the training dataset $L_{ref} = \frac{1}{K}\sum_{k=1}^K l(p(y \mid x_k, w), y_k)$, minimization procedure MINIMIZE, number of penalty iterations $M$, initial penalty parameter $c_0$, penalty parameter increase scheduler $\eta$, slack parameter $\gamma$, distance / divergence measure $D(\cdot, \cdot)$.

1: $\breve{w} \leftarrow w$; $\tilde{w} \leftarrow w$; $c \leftarrow c_0$
2: **for** $m \leftarrow 1$ to $M$ **do**
3:     $L_{pen} \leftarrow \frac{1}{K}\sum_{k=1}^K l(p(y \mid x_k, \tilde{w}), y_k) - (L_{ref} + \gamma)$
4:     $L_{adv} \leftarrow -D(p(y \mid x, w), p(y \mid x, \tilde{w}))$
5:     $L \leftarrow L_{adv} + c L_{pen}$
6:     $\tilde{w} \leftarrow \text{MINIMIZE}(L(\tilde{w}))$
7:     **if** $L_{adv}$ larger than all previous and $L_{pen} \leqslant 0$ **then**
8:         $\breve{w} \leftarrow \tilde{w}$
9:     $c \leftarrow \eta(c)$
10: **return** $\breve{w}$

---

**Practical Implementation.** Empirically, we found that directly executing the optimization procedure defined in Alg. 1 tends to result in adversarial models with similar predictive distribution for a given input across multiple searches. The vanilla implementation of Alg. 1 corresponds to an *untargeted* attack, known from the literature on adversarial attacks [Szegedy et al., 2013, Biggio et al., 2013]. To prevent the searches from converging to a single solution, we optimize the cross-entropy loss for one specific class during each search, which corresponds to a *targeted* attack. Each resulting adversarial model represents a local optimum of Eq. (7). We execute as many adversarial model searches as there are classes, dedicating one search to each class, unless otherwise specified. To compute Eq. (4), we use the predictive distributions $p(y \mid x, \tilde{w})$ of all models $\tilde{w}$ encountered during each penalty iteration of all searches, weighted by their posterior probability. The posterior probability is approximated with the negative exponential training loss, the likelihood, of models $\tilde{w}$. This approximate posterior probability is scaled with a temperature parameter, set as a hyperparameter. Further details are given in the appendix Sec. C.1.

## 4 Experiments

In this section, we compare previous uncertainty quantification methods and our method QUAM in a set of experiments. First, we assess the considered methods on a synthetic benchmark, on which it is feasible to compute a ground truth epistemic uncertainty. Then, we conduct challenging out-of-distribution (OOD) detection, adversarial example detection, misclassification detection and selective prediction experiments in the vision domain. We compare (1) QUAM, (2) cyclical Stochastic Gradient Hamiltonian Monte Carlo (cSG-HMC) [Zhang et al., 2020], (3) an efficient Laplace approximation (Laplace) [Daxberger et al., 2021], (4) MC dropout (MCD) [Gal and Ghahramani, 2016] and (5) Deep Ensembles (DE) [Lakshminarayanan et al., 2017] on their ability to estimate the epistemic uncertainty. Those baseline methods, especially Deep Ensembles, are persistently among the best performing uncertainty quantification methods across various benchmark tasks [Filos et al., 2019, Ovadia et al., 2019, Caldeira and Nord, 2020, Band et al., 2022]

### 4.1 Epistemic Uncertainty on Synthetic Dataset

We evaluated all considered methods on the two-moons dataset, created using the implementation of Pedregosa et al. [2011]. To obtain the ground truth uncertainty, we utilized Hamiltonian Monte Carlo (HMC) [Neal, 1996]. HMC is regarded as the most precise algorithm to approximate posterior expectations [Izmailov et al., 2021], but necessitates extreme computational expenses to be applied to models and datasets of practical scale. The results are depicted in Fig. 3. QUAM most closely matches the uncertainty estimate of the ground truth epistemic uncertainty obtained by HMC and excels especially on the regions further away from the decision boundary such as in the top left and bottom right of the plots. All other methods fail to capture the epistemic uncertainty in those regions as gradient descent on the training set fails to capture posterior modes with alternative predictive distributions in those parts and misses the important integral components. Experimental details and results for the epistemic uncertainty as in Eq. (2) are given in the appendix Sec. C.3.

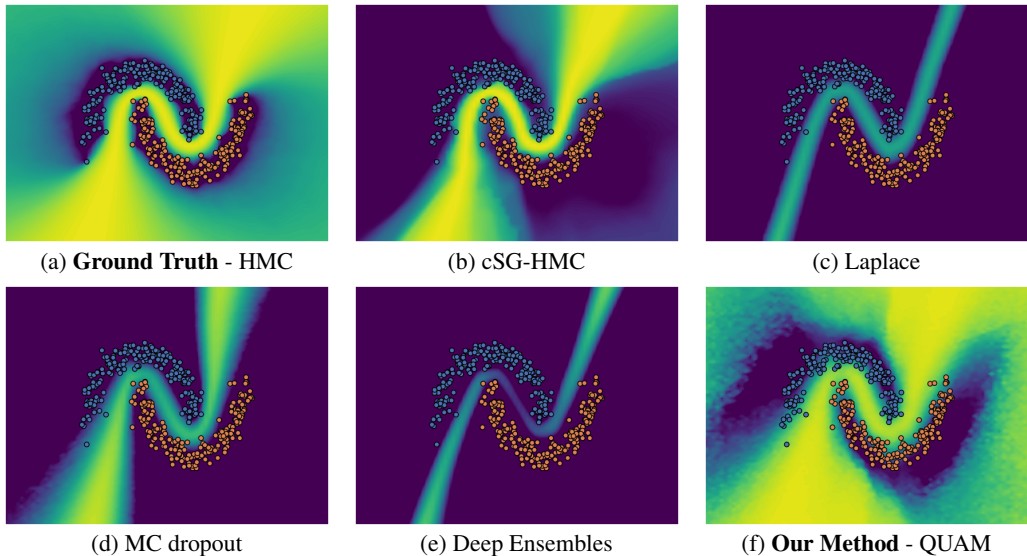

<table>
<tr><td>(a) Ground Truth - HMC</td><td>(b) cSG-HMC</td><td>(c) Laplace</td></tr>
<tr><td>(d) MC dropout</td><td>(e) Deep Ensembles</td><td>(f) Our Method - QUAM</td></tr>
</table>

Figure 3: Epistemic uncertainty as in Eq. (1) for two-moons. Yellow denotes high epistemic uncertainty. Purple denotes low epistemic uncertainty. HMC is considered as ground truth [Izmailov et al., 2021] and is most closely matched by QUAM. Artifacts for QUAM arise because it is applied to each test point individually, whereas other methods use the same sampled models for all test points.

## 4.2 Epistemic Uncertainty on Vision Datasets

We benchmark the ability of different methods to estimate the epistemic uncertainty of a given, pre-selected model (setting (b) as in Eq. (2)) in the context of (i) out-of-distribution (OOD) detection, (ii) adversarial example detection, (iii) misclassification detection and (iv) selective prediction. In all experiments, we assume to have access to a pre-trained model on the in-distribution (ID) training dataset, which we refer to as reference model. The epistemic uncertainty is expected to be higher for OOD samples, as they can be assigned to multiple ID classes, depending on the utilized features. Adversarial examples indicate that the model is misspecified on those inputs, thus we expect a higher epistemic uncertainty, the uncertainty about the model parameters. Furthermore, we expect higher epistemic uncertainty for misclassified samples than for correctly classified samples. Similarly, we expect the classifier to perform better on a subset of more certain samples. This is tested by evaluating the accuracy of the classifier on retained subsets of a certain fraction of samples with the lowest epistemic uncertainty [Filos et al., 2019, Band et al., 2022]. We report the AUROC for classifying the ID vs. OOD samples (i), the ID vs. the adversarial examples (ii), or the correctly classified vs. the misclassified samples (iii), using the epistemic uncertainty as score to distinguish the two classes respectively. For the selective prediction experiment (iv), we report the AUC of the accuracy vs. fraction of retained samples, using the epistemic uncertainty to determine the retained subsets.

**MNIST.** We perform OOD detection on the FMNIST [Xiao et al., 2017], KMNIST [Clanuwat et al., 2018], EMNIST [Cohen et al., 2017] and OMNIGLOT [Lake et al., 2015] test datasets as OOD datasets, using the LeNet [LeCun et al., 1998] architecture. The test dataset of MNIST [LeCun et al., 1998] is used as ID dataset. We utilize the aleatoric uncertainty of the reference model (as in Eq. (2)) as a baseline to assess the added value of estimating the epistemic uncertainty of the reference model. The results are listed in Tab. 1. QUAM outperforms all other methods on this task, with Deep Ensembles being the runner up method on all dataset pairs. Furthermore, we observed, that only the epistemic uncertainties obtained by Deep Ensembles and QUAM are able to surpass the performance of using the aleatoric uncertainty of the reference model.

**ImageNet-1K.** We conduct OOD detection, adversarial example detection, misclassification detection and selective prediction experiments on ImageNet-1K [Deng et al., 2009]. As OOD dataset, we use ImageNet-O [Hendrycks et al., 2021], which is a challenging OOD dataset that was explicitly created to be classified as an ID dataset with high confidence by conventional ImageNet-1K classifiers. Similarly, ImageNet-A [Hendrycks et al., 2021] is a dataset consisting of natural adversarial exam-

Table 1: MNIST results: AUROC using the epistemic uncertainty of a given, pre-selected model (as in Eq. (2)) as a score to distinguish between ID (MNIST) and OOD samples. We also report the AUROC when using the aleatoric uncertainty of the reference model (Reference).

| $\mathcal{D}_{ood}$ | Reference | cSG-HMC | Laplace | MCD | DE | QUAM |
|---|---|---|---|---|---|---|
| FMNIST | $.986_{\pm.005}$ | $.977_{\pm.004}$ | $.978_{\pm.004}$ | $.978_{\pm.005}$ | $.988_{\pm.001}$ | $\mathbf{.994}_{\pm.001}$ |
| KMNIST | $.966_{\pm.005}$ | $.957_{\pm.005}$ | $.959_{\pm.006}$ | $.956_{\pm.006}$ | $.990_{\pm.001}$ | $\mathbf{.994}_{\pm.001}$ |
| EMNIST | $.888_{\pm.007}$ | $.869_{\pm.012}$ | $.877_{\pm.011}$ | $.876_{\pm.008}$ | $.924_{\pm.003}$ | $\mathbf{.937}_{\pm.008}$ |
| OMNIGLOT | $.973_{\pm.003}$ | $.963_{\pm.004}$ | $.963_{\pm.003}$ | $.965_{\pm.003}$ | $.983_{\pm.001}$ | $\mathbf{.992}_{\pm.001}$ |

Table 2: ImageNet-1K results: AUROC using the epistemic uncertainty of a given, pre-selected model (as in Eq. (2)) to distinguish between ID (ImageNet-1K) and OOD samples. Furthermore, we report the AUROC when using the epistemic uncertainty for misclassification detection and the AUC of accuracy over fraction of retained predictions on the ImageNet-1K validation dataset. We also report results for all experiments, using the aleatoric uncertainty of the reference model (Reference).

| $\mathcal{D}_{ood}$ // Task | Reference | cSG-HMC | MCD | DE (LL) | DE (all) | QUAM |
|---|---|---|---|---|---|---|
| ImageNet-O | $.626_{\pm.004}$ | $.677_{\pm.005}$ | $.680_{\pm.003}$ | $.562_{\pm.004}$ | $.709_{\pm.005}$ | $\mathbf{.753}_{\pm.011}$ |
| ImageNet-A | $.792_{\pm.002}$ | $.799_{\pm.001}$ | $.827_{\pm.002}$ | $.686_{\pm.001}$ | $\mathbf{.874}_{\pm.004}$ | $.872_{\pm.003}$ |
| Misclassification | $.867_{\pm.007}$ | $.772_{\pm.011}$ | $.796_{\pm.014}$ | $.657_{\pm.009}$ | $.780_{\pm.009}$ | $\mathbf{.904}_{\pm.008}$ |
| Selective prediction | $.958_{\pm.003}$ | $.931_{\pm.003}$ | $.935_{\pm.006}$ | $.911_{\pm.004}$ | $.950_{\pm.002}$ | $\mathbf{.969}_{\pm.002}$ |

ples, which belong to the ID classes of ImageNet-1K, but are misclassified with high confidence by conventional ImageNet-1K classifiers. Furthermore, we evaluated the utility of the uncertainty score for misclassification detection of predictions of the reference model on the ImageNet-1K validation dataset. On the same dataset, we evaluated the accuracy of the reference model when only predicting on fractions of samples with the lowest epistemic uncertainty.

All ImageNet experiments were performed on variations of the EfficientNet architecture [Tan and Le, 2019]. Recent work by Kirichenko et al. [2022] showed that typical ImageNet-1K classifiers learn desired features of the data even if they rely on simple, spurious features for their prediction. Furthermore, they found, that last layer retraining on a dataset without the spurious correlation is sufficient to re-weight the importance that the classifier places on different features. This allows the classifier to ignore the spurious features and utilize the desired features for its prediction. Similarly, we apply QUAM on the last layer of the reference model. We compare against cSG-HMC applied to the last layer, MC dropout and Deep Ensembles. MC dropout was applied to the last layer as well, since the EfficientNet architectures utilize dropout only before the last layer. Two versions of Deep Ensembles were considered. First, Deep Ensembles aggregated from pre-trained EfficientNets of different network sizes (DE (all)). Second, Deep Ensembles of retrained last layers on the same encoder network (DE (LL)). We further utilize the aleatoric uncertainty of the reference model (as in Eq. (2)) as a baseline to assess the additional benefit of estimating the epistemic uncertainty of the reference model. The Laplace approximation was not feasible to compute on our hardware, even only for the last layer.

The results are listed in Tab. 2. Plots showing the respective curves of each experiment are depicted in Fig. C.8 in the appendix. We observe that using the epistemic uncertainty provided by DE (LL) has the worst performance throughout all experiments. While DE (all) performed second best on most tasks, MC dropout outperforms it on OOD detection on the ImageNet-O dataset. QUAM outperforms all other methods on all tasks we evaluated, except for ImageNet-A, where it performed on par with DE (all). Details about all experiments and additional results are given in the appendix Sec. C.4.

**Compute Efficiency.** As an ablation study, we investigate the performance of QUAM under a restricted computational budget. Therefore, the searches for adversarial models were performed on only a subset of classes instead of each eligible class, specifically the top $N$ most probable classes according to the predictive distribution of the given, pre-selected model. The computational budget between QUAM and MC dropout was matched by accounting for the number of forward pass equivalents required by each method. In this context, we assume that the backward pass corresponds to the computational cost of two forward passes. The results depicted in Fig. 4 show that QUAM

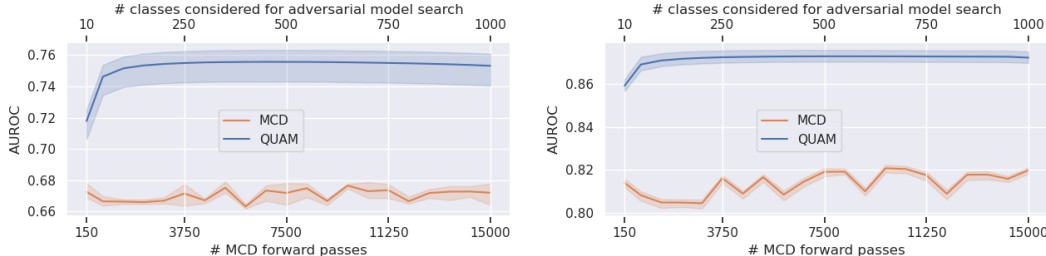

Figure 4: Inference speed vs. performance. MCD and QUAM evaluated on equal computational budget in terms of forward pass equivalents on ImageNet-O (left) and ImageNet-A (right) tasks.

outperforms MC dropout even under a very limited computational budget. Furthermore, training a single additional ensemble member for Deep Ensembles requires more compute than evaluating the entire ImageNet-O and ImageNet-A datasets with QUAM when performed on all 1000 classes.

## 5 Related Work

Quantifying predictive uncertainty, especially for deep learning models, is an active area of research. Classical uncertainty quantification methods such as Bayesian Neural Networks (BNNs) [MacKay, 1992, Neal, 1996] are challenging for deep learning, since (i) the Hessian or maximum-a-posterior (MAP) is difficult to estimate and (ii) regularization and normalization techniques cannot be treated [Antoráan et al., 2022]. Epistemic neural networks [Osband et al., 2021] add a variance term (the epinet) to the output only. Bayes By Backprop [Blundell et al., 2015] and variational neural networks [Oleksiienko et al., 2022] work only for small models as they require considerably more parameters. MC dropout [Gal and Ghahramani, 2016] casts applying dropout during inference as sampling from an approximate distribution. MC dropout was generalized to MC dropconnect [Mobiny et al., 2021]. Deep Ensembles [Lakshminarayanan et al., 2017] are often the best-performing uncertainty quantification method [Ovadia et al., 2019, Wursthorn et al., 2022]. Masksembles or Dropout Ensembles combine ensembling with MC dropout [Durasov et al., 2021]. Stochastic Weight Averaging approximates the posterior over the weights [Maddox et al., 2019]. Single forward pass methods are fast and they aim to capture a different notion of epistemic uncertainty through the distribution or distances of latent representations [Bradshaw et al., 2017, Liu et al., 2020, Mukhoti et al., 2021, van Amersfoort et al., 2021, Postels et al., 2021] rather than through posterior integrals. For further methods and a general overview of uncertainty estimation see e.g. Hüllermeier and Waegeman [2021], Abdar et al. [2021] and Gawlikowski et al. [2021].

## 6 Conclusion

We have introduced QUAM, a novel method that quantifies predictive uncertainty using adversarial models. Adversarial models identify important posterior modes that are missed by previous uncertainty quantification methods. We conducted various experiments on deep neural networks, for which epistemic uncertainty is challenging to estimate. On a synthetic dataset, we highlighted the strength of our method to capture epistemic uncertainty. Furthermore, we conducted experiments on large-scale benchmarks in the vision domain, where QUAM outperformed all previous methods.

Searching for adversarial models is computationally expensive and has to be done for each new test point. However, more efficient versions can be utilized. One can search for adversarial models while restricting the search to a subset of the parameters, e.g. to the last layer as was done for the ImageNet experiments, to the normalization parameters, or to the bias weights. Furthermore, there have been several advances for efficient fine-tuning of large models [Houlsby et al., 2019, Hu et al., 2021]. Utilizing those for more efficient versions of our algorithm is an interesting direction for future work.

Nevertheless, high stake applications justify this effort to obtain the best estimate of predictive uncertainty for each new test point. Furthermore, QUAM is applicable to quantify the predictive uncertainty of any single given model, regardless of whether uncertainty estimation was considered during the modeling process. This allows to assess the predictive uncertainty of foundation models or specialized models that are obtained externally.

## Acknowledgements

We would like to thank Angela Bitto-Nemling for providing relevant literature, organizing meetings, and giving feedback on this research project. Furthermore, we would like to thank Angela Bitto-Nemling, Daniel Klotz, and Sebastian Lehner for insightful discussions and provoking questions. The ELLIS Unit Linz, the LIT AI Lab, the Institute for Machine Learning, are supported by the Federal State Upper Austria. We thank the projects AI-MOTION (LIT-2018-6-YOU-212), DeepFlood (LIT-2019-8-YOU-213), Medical Cognitive Computing Center (MC3), INCONTROL-RL (FFG-881064), PRIMAL (FFG-873979), S3AI (FFG-872172), DL for GranularFlow (FFG-871302), EPILEPSIA (FFG-892171), AIRI FG 9-N (FWF-36284, FWF-36235), AI4GreenHeatingGrids(FFG- 899943), INTEGRATE (FFG-892418), ELISE (H2020-ICT-2019-3 ID: 951847), Stars4Waters (HORIZON-CL6-2021-CLIMATE-01-01). We thank Audi.JKU Deep Learning Center, TGW LOGISTICS GROUP GMBH, Silicon Austria Labs (SAL), FILL Gesellschaft mbH, Anyline GmbH, Google, ZF Friedrichshafen AG, Robert Bosch GmbH, UCB Biopharma SRL, Merck Healthcare KGaA, Verbund AG, GLS (Univ. Waterloo) Software Competence Center Hagenberg GmbH, TÜV Austria, Frauscher Sensonic, TRUMPF and the NVIDIA Corporation.

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

# Appendix

This is the appendix of the paper "**Quantification of Uncertainty with Adversarial Models**". It consists of three sections. In view of the increasing influence of contemporary machine learning research on the broader public, section A gives a societal impact statement. Following to this, section B gives details of our theoretical results, foremost about the measure of uncertainty used throughout our work. Furthermore, Mixture Importance Sampling for variance reduction is discussed. Finally, section C gives details about the experiments presented in the main paper, as well as further experiments.

# Contents of the Appendix

# List of Figures

## List of Tables

## A   Societal Impact Statement

In this work, we have focused on improving the predictive uncertainty estimation for machine learning models, specifically deep learning models. Our primary goal is to enhance the robustness and reliability of these predictions, which we believe have several positive societal impacts.

1. **Improved decision-making:** By providing more accurate predictive uncertainty estimates, we enable a broad range of stakeholders to make more informed decisions. This could have implications across various sectors, including healthcare, finance, and autonomous vehicles, where decision-making based on machine learning predictions can directly affect human lives and economic stability.

2. **Increased trust in machine learning systems:** By enhancing the reliability of machine learning models, our work may also contribute to increased public trust in these systems. This could foster greater acceptance and integration of machine learning technologies in everyday life, driving societal advancement.

3. **Promotion of responsible machine learning:** Accurate uncertainty estimation is crucial for the responsible deployment of machine learning systems. By advancing this area, our work promotes the use of those methods in an ethical, transparent, and accountable manner.

While we anticipate predominantly positive impacts, it is important to acknowledge potential negative impacts or challenges.

1. **Misinterpretation of uncertainty:** Even with improved uncertainty estimates, there is a risk that these might be misinterpreted or misused, potentially leading to incorrect decisions or unintended consequences. It is vital to couple advancements in this field with improved education and awareness around the interpretation of uncertainty in AI systems.

2. **Increased reliance on machine learning systems:** While increased trust in machine learning systems is beneficial, there is a risk it could lead to over-reliance on these systems, potentially resulting in reduced human oversight or critical thinking. It's important that robustness and reliability improvements don't result in blind trust.

3. **Inequitable distribution of benefits:** As with any technological advancement, there is a risk that the benefits might not be evenly distributed, potentially exacerbating existing societal inequalities. We urge policymakers and practitioners to consider this when implementing our findings.

In conclusion, while our work aims to make significant positive contributions to society, we believe it is essential to consider these potential negative impacts and take steps to mitigate them proactively.

# B Theoretical Results

## B.1 Measuring Predictive Uncertainty

In this section, we first discuss the usage of the entropy and the cross-entropy as measures of predictive uncertainty. Following this, we introduce the two settings (a) and (b) (see Sec. 2) in detail for the predictive distributions of probabilistic models in classification and regression. Finally, we discuss Mixture Importance Sampling for variance reduction of the uncertainty estimator.

### B.1.1 Entropy and Cross-Entropy as Measures of Predictive Uncertainty

Shannon and Elwood [1948] defines the entropy $\mathrm{H}[\boldsymbol{p}] = -\sum_{i=1}^{N} p_i \log p_i$ as a measure of the amount of uncertainty of a discrete probability distribution $\boldsymbol{p} = (p_1, \ldots, p_N)$ and states that it measures how much "choice" is involved in the selection of a class $i$. See also Jaynes [1957], Cover and Thomas [2006] for an elaboration on this topic. The value $-\log p_i$ has been called "surprisal" [Tribus, 1961] (page 64, Subsection 2.9.1) and has been used in computational linguistics [Hale, 2001]. Hence, the entropy is the expected or mean surprisal. Instead of "surprisal" also the terms "information content", "self-information", or "Shannon information" are used.

The cross-entropy $\mathrm{CE}[\boldsymbol{p}, \boldsymbol{q}] = -\sum_{i=1}^{N} p_i \log q_i$ between two discrete probability distributions $\boldsymbol{p} = (p_1, \ldots, p_N)$ and $\boldsymbol{q} = (q_1, \ldots, q_N)$ measures the expectation of the surprisal of $\boldsymbol{q}$ under distribution $\boldsymbol{p}$. Like the entropy, the cross-entropy is a mean of surprisals, therefore can be considered as a measure to quantify uncertainty. The higher surprisals are on average, the higher the uncertainty. The cross-entropy has increased uncertainty compared to the entropy since more surprising events are expected when selecting events via $\boldsymbol{p}$ instead of $\boldsymbol{q}$. Only if those distributions coincide, there is no additional surprisal and the cross-entropy is equal to the entropy of the distributions. The cross-entropy depends on the uncertainty of the two distributions and how different they are. In particular, high surprisal of $q_i$ and low surprisal of $p_i$ strongly increase the cross-entropy since unexpected events are more frequent, that is, we are more often surprised. Thus, the cross-entropy does not only measure the uncertainty under distribution $\boldsymbol{p}$, but also the difference of the distributions. The average surprisal via the cross-entropy depends on the uncertainty of $\boldsymbol{p}$ and the difference between $\boldsymbol{p}$ and $\boldsymbol{q}$:

$$\mathrm{CE}[\boldsymbol{p}, \boldsymbol{q}] = -\sum_{i=1}^{N} p_i \log q_i \tag{9}$$

$$= -\sum_{i=1}^{N} p_i \log p_i + \sum_{i=1}^{N} p_i \log \frac{p_i}{q_i}$$

$$= \mathrm{H}[\boldsymbol{p}] + \mathrm{D_{KL}}(\boldsymbol{p} \,\|\, \boldsymbol{q}) ,$$

where the Kullback-Leibler divergence $\mathrm{D_{KL}}(\cdot \,\|\, \cdot)$ is

$$\mathrm{D_{KL}}(\boldsymbol{p} \,\|\, \boldsymbol{q}) = \sum_{i=1}^{N} p_i \log \frac{p_i}{q_i} . \tag{10}$$

The Kullback-Leibler divergence measures the difference in the distributions via their average difference of surprisals. Furthermore, it measures the decrease in uncertainty when shifting from the estimate $\boldsymbol{p}$ to the true $\boldsymbol{q}$ [Seidenfeld, 1986, Adler et al., 2008].

Therefore, the cross-entropy can serve to measure the total uncertainty, where the entropy is used as aleatoric uncertainty and the difference of distributions is used as the epistemic uncertainty. We assume that $\boldsymbol{q}$ is the true distribution that is estimated by the distribution $\boldsymbol{p}$. We quantify the total uncertainty of $\boldsymbol{p}$ as the sum of the entropy of $\boldsymbol{p}$ (aleatoric uncertainty) and the Kullback-Leibler divergence to $\boldsymbol{q}$ (epistemic uncertainty). In accordance with Apostolakis [1991] and Helton [1997], the aleatoric uncertainty measures the stochasticity of sampling from $\boldsymbol{p}$, while the epistemic uncertainty measures the deviation of the parameters $\boldsymbol{p}$ from the true parameters $\boldsymbol{q}$.

In the context of quantifying uncertainty through probability distributions, other measures such as the variance have been proposed [Zidek and vanEeden, 2003]. For uncertainty estimation in the context of deep learning systems, e.g. Gal [2016], Kendall and Gal [2017], Depeweg et al. [2018] proposed to use the variance of the BMA predictive distribution as a measure of uncertainty. Entropy and variance capture different notions of uncertainty and investigating measures based on the variance of the predictive distribution is an interesting avenue for future work.

### B.1.2  Classification

**Setting (a): Expected uncertainty when selecting a model.**  We assume to have training data $\mathcal{D}$ and an input $\boldsymbol{x}$. We want to know the uncertainty in predicting a class $\boldsymbol{y}$ from $\boldsymbol{x}$ when we first choose a model $\tilde{\boldsymbol{w}}$ based on the posterior $p(\tilde{\boldsymbol{w}} \mid \mathcal{D})$ an then use the chosen model $\tilde{\boldsymbol{w}}$ to choose a class for input $\boldsymbol{x}$ according to the predictive distribution $p(\boldsymbol{y} \mid \boldsymbol{x}, \tilde{\boldsymbol{w}})$. The uncertainty in predicting the class arises from choosing a model (epistemic) and from choosing a class using this probabilistic model (aleatoric).

Through Bayesian model averaging, we obtain the following probability of selecting a class:

$$p(\boldsymbol{y} \mid \boldsymbol{x}, \mathcal{D}) \; = \; \int_{\mathcal{W}} p(\boldsymbol{y} \mid \boldsymbol{x}, \tilde{\boldsymbol{w}}) \, p(\tilde{\boldsymbol{w}} \mid \mathcal{D}) \, \mathrm{d}\tilde{\boldsymbol{w}} \, . \tag{11}$$

The total uncertainty is commonly measured as the entropy of this probability distribution [Houlsby et al., 2011, Gal, 2016, Depeweg et al., 2018, Hüllermeier and Waegeman, 2021]:

$$\mathrm{H}[p(\boldsymbol{y} \mid \boldsymbol{x}, \mathcal{D})] \, . \tag{12}$$

We can reformulate the total uncertainty as the expected cross-entropy:

$$\begin{aligned}
\mathrm{H}[p(\boldsymbol{y} \mid \boldsymbol{x}, \mathcal{D})] \; &= \; - \sum_{\boldsymbol{y} \in \mathcal{Y}} p(\boldsymbol{y} \mid \boldsymbol{x}, \mathcal{D}) \, \log p(\boldsymbol{y} \mid \boldsymbol{x}, \mathcal{D}) \\
&= \; - \sum_{\boldsymbol{y} \in \mathcal{Y}} \log p(\boldsymbol{y} \mid \boldsymbol{x}, \mathcal{D}) \int_{\mathcal{W}} p(\boldsymbol{y} \mid \boldsymbol{x}, \tilde{\boldsymbol{w}}) \, p(\tilde{\boldsymbol{w}} \mid \mathcal{D}) \, \mathrm{d}\tilde{\boldsymbol{w}} \\
&= \; \int_{\mathcal{W}} \left( - \sum_{\boldsymbol{y} \in \mathcal{Y}} p(\boldsymbol{y} \mid \boldsymbol{x}, \tilde{\boldsymbol{w}}) \, \log p(\boldsymbol{y} \mid \boldsymbol{x}, \mathcal{D}) \right) p(\tilde{\boldsymbol{w}} \mid \mathcal{D}) \, \mathrm{d}\tilde{\boldsymbol{w}} \\
&= \; \int_{\mathcal{W}} \mathrm{CE}[p(\boldsymbol{y} \mid \boldsymbol{x}, \tilde{\boldsymbol{w}}) \, , \, p(\boldsymbol{y} \mid \boldsymbol{x}, \mathcal{D})] \, p(\tilde{\boldsymbol{w}} \mid \mathcal{D}) \, \mathrm{d}\tilde{\boldsymbol{w}} \, .
\end{aligned} \tag{13}$$

We can split the total uncertainty into the aleatoric and epistemic uncertainty [Houlsby et al., 2011, Gal, 2016, Smith and Gal, 2018]:

$$\begin{aligned}
\int_{\mathcal{W}} &\mathrm{CE}[p(\boldsymbol{y} \mid \boldsymbol{x}, \tilde{\boldsymbol{w}}) \, , \, p(\boldsymbol{y} \mid \boldsymbol{x}, \mathcal{D})] \, p(\tilde{\boldsymbol{w}} \mid \mathcal{D}) \, \mathrm{d}\tilde{\boldsymbol{w}} \\
&= \; \int_{\mathcal{W}} \left( \mathrm{H}[p(\boldsymbol{y} \mid \boldsymbol{x}, \tilde{\boldsymbol{w}})] \, + \, \mathrm{D}_{\mathrm{KL}}(p(\boldsymbol{y} \mid \boldsymbol{x}, \tilde{\boldsymbol{w}}) \, \| \, p(\boldsymbol{y} \mid \boldsymbol{x}, \mathcal{D}))) \, \right) p(\tilde{\boldsymbol{w}} \mid \mathcal{D}) \, \mathrm{d}\tilde{\boldsymbol{w}} \\
&= \; \int_{\mathcal{W}} \mathrm{H}[p(\boldsymbol{y} \mid \boldsymbol{x}, \tilde{\boldsymbol{w}})] \, p(\tilde{\boldsymbol{w}} \mid \mathcal{D}) \, \mathrm{d}\tilde{\boldsymbol{w}} \, + \, \int_{\mathcal{W}} \mathrm{D}_{\mathrm{KL}}(p(\boldsymbol{y} \mid \boldsymbol{x}, \tilde{\boldsymbol{w}}) \, \| \, p(\boldsymbol{y} \mid \boldsymbol{x}, \mathcal{D})) \, p(\tilde{\boldsymbol{w}} \mid \mathcal{D}) \, \mathrm{d}\tilde{\boldsymbol{w}} \\
&= \; \int_{\mathcal{W}} \mathrm{H}[p(\boldsymbol{y} \mid \boldsymbol{x}, \tilde{\boldsymbol{w}})] \, p(\tilde{\boldsymbol{w}} \mid \mathcal{D}) \, \mathrm{d}\tilde{\boldsymbol{w}} \, + \, \mathrm{I}[Y \, ; \, W \mid \boldsymbol{x}, \mathcal{D}] \, .
\end{aligned} \tag{14}$$

We verify the last equality in Eq. (14), i.e. that the Mutual Information is equal to the expected Kullback-Leibler divergence:

$$\begin{aligned}
\mathrm{I}[Y \, ; \, W \mid \boldsymbol{x}, \mathcal{D}] \; &= \; \int_{\mathcal{W}} \sum_{\boldsymbol{y} \in \mathcal{Y}} p(\boldsymbol{y}, \tilde{\boldsymbol{w}} \mid \boldsymbol{x}, \mathcal{D}) \, \log \frac{p(\boldsymbol{y}, \tilde{\boldsymbol{w}} \mid \boldsymbol{x}, \mathcal{D})}{p(\boldsymbol{y} \mid \boldsymbol{x}, \mathcal{D}) \, p(\tilde{\boldsymbol{w}} \mid \mathcal{D})} \, \mathrm{d}\tilde{\boldsymbol{w}} \\
&= \; \int_{\mathcal{W}} \sum_{\boldsymbol{y} \in \mathcal{Y}} p(\boldsymbol{y} \mid \boldsymbol{x}, \tilde{\boldsymbol{w}}) \, p(\tilde{\boldsymbol{w}} \mid \mathcal{D}) \, \log \frac{p(\boldsymbol{y} \mid \boldsymbol{x}, \tilde{\boldsymbol{w}}) \, p(\tilde{\boldsymbol{w}} \mid \mathcal{D})}{p(\boldsymbol{y} \mid \boldsymbol{x}, \mathcal{D}) \, p(\tilde{\boldsymbol{w}} \mid \mathcal{D})} \, \mathrm{d}\tilde{\boldsymbol{w}} \\
&= \; \int_{\mathcal{W}} \sum_{\boldsymbol{y} \in \mathcal{Y}} p(\boldsymbol{y} \mid \boldsymbol{x}, \tilde{\boldsymbol{w}}) \, \log \frac{p(\boldsymbol{y} \mid \boldsymbol{x}, \tilde{\boldsymbol{w}})}{p(\boldsymbol{y} \mid \boldsymbol{x}, \mathcal{D})} \, p(\tilde{\boldsymbol{w}} \mid \mathcal{D}) \, \mathrm{d}\tilde{\boldsymbol{w}} \\
&= \; \int_{\mathcal{W}} \mathrm{D}_{\mathrm{KL}}(p(\boldsymbol{y} \mid \boldsymbol{x}, \tilde{\boldsymbol{w}}) \, \| \, p(\boldsymbol{y} \mid \boldsymbol{x}, \mathcal{D})) \, p(\tilde{\boldsymbol{w}} \mid \mathcal{D}) \, \mathrm{d}\tilde{\boldsymbol{w}} \, .
\end{aligned} \tag{15}$$

This is possible because the label is dependent on the selected model. First, a model is selected, then a label is chosen with the selected model. To summarize, the predictive uncertainty is measured by:

$$\mathrm{H}[p(\boldsymbol{y} \mid \boldsymbol{x}, \mathcal{D})] = \int_{\mathcal{W}} \mathrm{H}[p(\boldsymbol{y} \mid \boldsymbol{x}, \tilde{\boldsymbol{w}})] \, p(\tilde{\boldsymbol{w}} \mid \mathcal{D}) \, \mathrm{d}\tilde{\boldsymbol{w}} \; + \; \mathrm{I}[Y \, ; \, W \mid \boldsymbol{x}, \mathcal{D}] \tag{16}$$

$$= \int_{\mathcal{W}} \mathrm{H}[p(\boldsymbol{y} \mid \boldsymbol{x}, \tilde{\boldsymbol{w}})] \, p(\tilde{\boldsymbol{w}} \mid \mathcal{D}) \, \mathrm{d}\tilde{\boldsymbol{w}}$$

$$+ \int_{\mathcal{W}} \mathrm{D}_{\mathrm{KL}}(p(\boldsymbol{y} \mid \boldsymbol{x}, \tilde{\boldsymbol{w}}) \, \| \, p(\boldsymbol{y} \mid \boldsymbol{x}, \mathcal{D})) \, p(\tilde{\boldsymbol{w}} \mid \mathcal{D}) \, \mathrm{d}\tilde{\boldsymbol{w}}$$

$$= \int_{\mathcal{W}} \mathrm{CE}[p(\boldsymbol{y} \mid \boldsymbol{x}, \tilde{\boldsymbol{w}}) \, , \, p(\boldsymbol{y} \mid \boldsymbol{x}, \mathcal{D})] \, p(\tilde{\boldsymbol{w}} \mid \mathcal{D}) \, \mathrm{d}\tilde{\boldsymbol{w}} \; .$$

The total uncertainty is given by the entropy of the Bayesian model average predictive distribution, which we showed is equal to the expected cross-entropy between the predictive distributions of candidate models $\tilde{\boldsymbol{w}}$ selected according to the posterior and the Bayesian model average predictive distribution. The aleatoric uncertainty is the expected entropy of candidate models drawn from the posterior, which can also be interpreted as the entropy we expect when selecting a model according to the posterior. Therefore, if all models likely under the posterior have low surprisal, the aleatoric uncertainty in this setting is low. The epistemic uncertainty is the expected KL divergence between the the predictive distributions of candidate models and the Bayesian model average predictive distribution. Therefore, if all models likely under the posterior have low divergence of their predictive distribution to the Bayesian model average predictive distribution, the epistemic uncertainty in this setting is low.

**Setting (b): Uncertainty of a given, pre-selected model.** We assume to have training data $\mathcal{D}$, an input $\boldsymbol{x}$, and a given, pre-selected model with parameters $\boldsymbol{w}$ and predictive distribution $p(\boldsymbol{y} \mid \boldsymbol{x}, \boldsymbol{w})$. Using the predictive distribution of the model, a class $\boldsymbol{y}$ is selected based on $\boldsymbol{x}$, therefore there is uncertainty about which $\boldsymbol{y}$ is selected. Furthermore, we assume that the true model with predictive distribution $p(\boldsymbol{y} \mid \boldsymbol{x}, \boldsymbol{w}^*)$ and parameters $\boldsymbol{w}^*$ has generated the training data $\mathcal{D}$ and will also generate the observed (real world) $\boldsymbol{y}^*$ from $\boldsymbol{x}$ that we want to predict. The true model is only revealed later, e.g. via more samples or by receiving knowledge about $\boldsymbol{w}^*$. Hence, there is uncertainty about the parameters of the true model. Revealing the true model is viewed as drawing a true model from all possible true models according to their agreement with $\mathcal{D}$. Note, to reveal the true model is not necessary in our framework but helpful for the intuition of drawing a true model. We neither consider uncertainty about the model class nor the modeling nor about the training data. In summary, there is uncertainty about drawing a class from the predictive distribution of the given, pre-selected model and uncertainty about drawing the true parameters of the model distribution.

According to Apostolakis [1991] and Helton [1997], the aleatoric uncertainty is the variability of selecting a class $\boldsymbol{y}$ via $p(\boldsymbol{y} \mid \boldsymbol{x}, \boldsymbol{w})$. Using the entropy, the aleatoric uncertainty is

$$\mathrm{H}[p(\boldsymbol{y} \mid \boldsymbol{x}, \boldsymbol{w})] \; . \tag{17}$$

Also according to Apostolakis [1991] and Helton [1997], the epistemic uncertainty is the uncertainty about the parameters $\boldsymbol{w}$ of the distribution, that is, a difference measure between $\boldsymbol{w}$ and the true parameters $\boldsymbol{w}^*$. We use as a measure for the epistemic uncertainty the Kullback-Leibler divergence:

$$\mathrm{D}_{\mathrm{KL}}(p(\boldsymbol{y} \mid \boldsymbol{x}, \boldsymbol{w}) \, \| \, p(\boldsymbol{y} \mid \boldsymbol{x}, \boldsymbol{w}^*)) \; . \tag{18}$$

The total uncertainty is the aleatoric uncertainty plus the epistemic uncertainty, which is the cross-entropy between $p(\boldsymbol{y} \mid \boldsymbol{x}, \boldsymbol{w})$ and $p(\boldsymbol{y} \mid \boldsymbol{x}, \boldsymbol{w}^*)$:

$$\mathrm{CE}[p(\boldsymbol{y} \mid \boldsymbol{x}, \boldsymbol{w}) \, , \, p(\boldsymbol{y} \mid \boldsymbol{x}, \boldsymbol{w}^*)] \; = \; \mathrm{H}[p(\boldsymbol{y} \mid \boldsymbol{x}, \boldsymbol{w})] \; + \; \mathrm{D}_{\mathrm{KL}}(p(\boldsymbol{y} \mid \boldsymbol{x}, \boldsymbol{w}) \, \| \, p(\boldsymbol{y} \mid \boldsymbol{x}, \boldsymbol{w}^*)) \; . \tag{19}$$

However, we do not know the true parameters $\boldsymbol{w}^*$. The posterior $p(\tilde{\boldsymbol{w}} \mid \mathcal{D})$ gives us the likelihood of $\tilde{\boldsymbol{w}}$ being the true parameters $\boldsymbol{w}^*$. We assume that the true model is revealed later. Therefore we use the expected Kullback-Leibler divergence for the epistemic uncertainty:

$$\int_{\mathcal{W}} \mathrm{D}_{\mathrm{KL}}(p(\boldsymbol{y} \mid \boldsymbol{x}, \boldsymbol{w}) \, \| \, p(\boldsymbol{y} \mid \boldsymbol{x}, \tilde{\boldsymbol{w}})) \, p(\tilde{\boldsymbol{w}} \mid \mathcal{D}) \, \mathrm{d}\tilde{\boldsymbol{w}} \; . \tag{20}$$

Consequently, the total uncertainty is

$$\mathrm{H}[p(\boldsymbol{y} \mid \boldsymbol{x}, \boldsymbol{w})] \; + \; \int_{\mathcal{W}} \mathrm{D}_{\mathrm{KL}}(p(\boldsymbol{y} \mid \boldsymbol{x}, \boldsymbol{w}) \, \| \, p(\boldsymbol{y} \mid \boldsymbol{x}, \tilde{\boldsymbol{w}})) \, p(\tilde{\boldsymbol{w}} \mid \mathcal{D}) \, \mathrm{d}\tilde{\boldsymbol{w}} \; . \tag{21}$$

The total uncertainty can therefore be expressed by the expected cross-entropy as it was in setting (a) (see Eq. (16)), but between $p(\boldsymbol{y} \mid \boldsymbol{x}, \boldsymbol{w})$ and $p(\boldsymbol{y} \mid \boldsymbol{x}, \tilde{\boldsymbol{w}})$:

$$\int_{\mathcal{W}} \mathrm{CE}[p(\boldsymbol{y} \mid \boldsymbol{x}, \boldsymbol{w}), p(\boldsymbol{y} \mid \boldsymbol{x}, \tilde{\boldsymbol{w}})] \, p(\tilde{\boldsymbol{w}} \mid \mathcal{D}) \, \mathrm{d}\tilde{\boldsymbol{w}} \tag{22}$$

$$= \int_{\mathcal{W}} (\mathrm{H}[p(\boldsymbol{y} \mid \boldsymbol{x}, \boldsymbol{w})] + \mathrm{D}_{\mathrm{KL}}(p(\boldsymbol{y} \mid \boldsymbol{x}, \boldsymbol{w}) \,\|\, p(\boldsymbol{y} \mid \boldsymbol{x}, \tilde{\boldsymbol{w}}))) \, p(\tilde{\boldsymbol{w}} \mid \mathcal{D}) \, \mathrm{d}\tilde{\boldsymbol{w}}$$

$$= \mathrm{H}[p(\boldsymbol{y} \mid \boldsymbol{x}, \boldsymbol{w})] + \int_{\mathcal{W}} \mathrm{D}_{\mathrm{KL}}(p(\boldsymbol{y} \mid \boldsymbol{x}, \boldsymbol{w}) \,\|\, p(\boldsymbol{y} \mid \boldsymbol{x}, \tilde{\boldsymbol{w}})) \, p(\tilde{\boldsymbol{w}} \mid \mathcal{D}) \, \mathrm{d}\tilde{\boldsymbol{w}} \,.$$

### B.1.3 Regression

We follow Depeweg et al. [2018] and measure the predictive uncertainty in a regression setting using the differential entropy $\mathrm{H}[p(y \mid \boldsymbol{x}, \boldsymbol{w})] = -\int_{\mathcal{Y}} p(y \mid \boldsymbol{x}, \boldsymbol{w}) \log p(y \mid \boldsymbol{x}, \boldsymbol{w}) \mathrm{d}y$ of the predictive distribution $p(y \mid \boldsymbol{x}, \boldsymbol{w})$ of a probabilistic model. In the following, we assume that we are modeling a Gaussian distribution, but other continuous probability distributions e.g. a Laplace lead to similar results. The model thus has to provide estimators for the mean $\mu(\boldsymbol{x}, \boldsymbol{w})$ and variance $\sigma^2(\boldsymbol{x}, \boldsymbol{w})$ of the Gaussian. The predictive distribution is given by

$$p(y \mid \boldsymbol{x}, \boldsymbol{w}) = (2\pi \, \sigma^2(\boldsymbol{x}, \boldsymbol{w}))^{-\frac{1}{2}} \exp\left\{ -\frac{(y - \mu(\boldsymbol{x}, \boldsymbol{w}))^2}{2 \, \sigma^2(\boldsymbol{x}, \boldsymbol{w})} \right\}. \tag{23}$$

The differential entropy of a Gaussian distribution is given by

$$\mathrm{H}[p(y \mid \boldsymbol{x}, \boldsymbol{w})] = -\int_{\mathcal{Y}} p(y \mid \boldsymbol{x}, \boldsymbol{w}) \log p(y \mid \boldsymbol{x}, \boldsymbol{w}) \, \mathrm{d}y \tag{24}$$

$$= \frac{1}{2} \, \log(\sigma^2(\boldsymbol{x}, \boldsymbol{w})) + \log(2\pi) + \frac{1}{2} \,.$$

The KL divergence between two Gaussian distributions is given by

$$\mathrm{D}_{\mathrm{KL}}(p(y \mid \boldsymbol{x}, \boldsymbol{w}) \,\|\, p(y \mid \boldsymbol{x}, \tilde{\boldsymbol{w}})) \tag{25}$$

$$= -\int_{\mathcal{Y}} p(y \mid \boldsymbol{x}, \boldsymbol{w}) \log\left( \frac{p(y \mid \boldsymbol{x}, \boldsymbol{w})}{p(y \mid \boldsymbol{x}, \tilde{\boldsymbol{w}})} \right) \, \mathrm{d}y$$

$$= \frac{1}{2} \, \log\left( \frac{\sigma^2(\boldsymbol{x}, \tilde{\boldsymbol{w}})}{\sigma^2(\boldsymbol{x}, \boldsymbol{w})} \right) + \frac{\sigma^2(\boldsymbol{x}, \boldsymbol{w}) + (\mu(\boldsymbol{x}, \boldsymbol{w}) - \mu(\boldsymbol{x}, \tilde{\boldsymbol{w}}))^2}{2 \, \sigma^2(\boldsymbol{x}, \tilde{\boldsymbol{w}})} - \frac{1}{2} \,.$$

**Setting (a): Expected uncertainty when selecting a model.** Depeweg et al. [2018] consider the differential entropy of the Bayesian model average $p(y \mid \boldsymbol{x}, \mathcal{D}) = \int_W p(y \mid \boldsymbol{x}, \tilde{\boldsymbol{w}}) p(\tilde{\boldsymbol{w}} \mid \mathcal{D}) \mathrm{d}\tilde{\boldsymbol{w}}$, which is equal to the expected cross-entropy and can be decomposed into the expected differential entropy and Kullback-Leibler divergence. Therefore, the expected uncertainty when selecting a model is given by

$$\int_{\mathcal{W}} \mathrm{CE}[p(y \mid \boldsymbol{x}, \tilde{\boldsymbol{w}}), p(y \mid \boldsymbol{x}, \mathcal{D})] \, p(\tilde{\boldsymbol{w}} \mid \mathcal{D}) \, \mathrm{d}\tilde{\boldsymbol{w}} = \mathrm{H}[p(y \mid \boldsymbol{x}, \mathcal{D})] \tag{26}$$

$$= \int_{\mathcal{W}} \mathrm{H}[p(y \mid \boldsymbol{x}, \tilde{\boldsymbol{w}})] \, p(\tilde{\boldsymbol{w}} \mid \mathcal{D}) \, \mathrm{d}\tilde{\boldsymbol{w}} + \int_{\mathcal{W}} \mathrm{D}_{\mathrm{KL}}(p(y \mid \boldsymbol{x}, \tilde{\boldsymbol{w}}) \,\|\, p(y \mid \boldsymbol{x}, \mathcal{D})) \, p(\tilde{\boldsymbol{w}} \mid \mathcal{D}) \, \mathrm{d}\tilde{\boldsymbol{w}}$$

$$= \int_{\mathcal{W}} \frac{1}{2} \, \log(\sigma^2(\boldsymbol{x}, \tilde{\boldsymbol{w}})) \, p(\tilde{\boldsymbol{w}} \mid \mathcal{D}) \, \mathrm{d}\tilde{\boldsymbol{w}} + \log(2\pi)$$

$$+ \int_{\mathcal{W}} \mathrm{D}_{\mathrm{KL}}(p(y \mid \boldsymbol{x}, \tilde{\boldsymbol{w}}) \,\|\, p(y \mid \boldsymbol{x}, \mathcal{D})) \, p(\tilde{\boldsymbol{w}} \mid \mathcal{D}) \, \mathrm{d}\tilde{\boldsymbol{w}} \,.$$

**Setting (b): Uncertainty of a given, pre-selected model.** Synonymous to the classification setting, the uncertainty of a given, pre-selected model $\boldsymbol{w}$ is given by

$$\int_{\mathcal{W}} \mathrm{CE}[p(y \mid \boldsymbol{x}, \boldsymbol{w}), \, p(y \mid \boldsymbol{x}, \tilde{\boldsymbol{w}})] \, p(\tilde{\boldsymbol{w}} \mid \mathcal{D}) \, \mathrm{d}\tilde{\boldsymbol{w}} \tag{27}$$

$$= \mathrm{H}[p(\boldsymbol{y} \mid \boldsymbol{x}, \boldsymbol{w})] \, + \, \int_{\mathcal{W}} \mathrm{D}_{\mathrm{KL}}(p(\boldsymbol{y} \mid \boldsymbol{x}, \boldsymbol{w}) \, \| \, p(\boldsymbol{y} \mid \boldsymbol{x}, \tilde{\boldsymbol{w}})) \, p(\tilde{\boldsymbol{w}} \mid \mathcal{D}) \, \mathrm{d}\tilde{\boldsymbol{w}}$$

$$= \frac{1}{2} \, \log(\sigma^2(\boldsymbol{x}, \boldsymbol{w})) \, + \, \log(2\pi)$$

$$+ \int_{\mathcal{W}} \frac{1}{2} \, \log\left(\frac{\sigma^2(\boldsymbol{x}, \tilde{\boldsymbol{w}})}{\sigma^2(\boldsymbol{x}, \boldsymbol{w})}\right) \, + \, \frac{\sigma^2(\boldsymbol{x}, \boldsymbol{w}) \, + \, (\mu(\boldsymbol{x}, \boldsymbol{w}) \, - \, \mu(\boldsymbol{x}, \tilde{\boldsymbol{w}}))^2}{2 \, \sigma^2(\boldsymbol{x}, \tilde{\boldsymbol{w}})} \, p(\tilde{\boldsymbol{w}} \mid \mathcal{D}) \, \mathrm{d}\tilde{\boldsymbol{w}} \, .$$

**Homoscedastic, Model Invariant Noise.** We assume, that noise is homoscedastic for all inputs $\boldsymbol{x} \in \mathcal{X}$, thus $\sigma^2(\boldsymbol{x}, \boldsymbol{w}) = \sigma^2(\boldsymbol{w})$. Furthermore, most models in regression do not explicitly model the variance in their training objective. For such a model $\boldsymbol{w}$, we can estimate the variance on a validation dataset $\mathcal{D}_{\mathrm{val}} = \{(\boldsymbol{x}_n, y_n)\}|_{n=1}^N$ as

$$\hat{\sigma}^2(\boldsymbol{w}) = \frac{1}{N} \, \sum_{n=1}^N \, (y_n - \mu(\boldsymbol{x}_n, \boldsymbol{w}))^2 \, . \tag{28}$$

If we assume that all reasonable models under the posterior will have similar variances ($\hat{\sigma}^2(\boldsymbol{w}) \approx \sigma^2(\tilde{\boldsymbol{w}})$ for $\tilde{\boldsymbol{w}} \sim p(\tilde{\boldsymbol{w}} \mid \mathcal{D})$), the uncertainty of a prediction using the given, pre-selected model $\boldsymbol{w}$ is given by

$$\int_{\mathcal{W}} \mathrm{CE}[p(y \mid \boldsymbol{x}, \boldsymbol{w}), \, p(y \mid \boldsymbol{x}, \tilde{\boldsymbol{w}})] \, p(\tilde{\boldsymbol{w}} \mid \mathcal{D}) \, \mathrm{d}\tilde{\boldsymbol{w}} \tag{29}$$

$$\approx \frac{1}{2} \, \log(\hat{\sigma}^2(\boldsymbol{w})) \, + \, \log(2\pi)$$

$$+ \int_{\mathcal{W}} \frac{1}{2} \, \log\left(\frac{\hat{\sigma}^2(\boldsymbol{w})}{\hat{\sigma}^2(\boldsymbol{w})}\right) \, + \, \frac{\hat{\sigma}^2(\boldsymbol{w}) \, + \, (\mu(\boldsymbol{x}, \boldsymbol{w}) \, - \, \mu(\boldsymbol{x}, \tilde{\boldsymbol{w}}))^2}{2 \, \hat{\sigma}^2(\boldsymbol{w})} \, p(\tilde{\boldsymbol{w}} \mid \mathcal{D}) \, \mathrm{d}\tilde{\boldsymbol{w}}$$

$$= \frac{1}{2} \, \log(\hat{\sigma}^2(\boldsymbol{w})) \, + \, \frac{1}{\hat{\sigma}^2(\boldsymbol{w})} \, \int_{\mathcal{W}} \, (\mu(\boldsymbol{x}, \boldsymbol{w}) \, - \, \mu(\boldsymbol{x}, \tilde{\boldsymbol{w}}))^2 \, p(\tilde{\boldsymbol{w}} \mid \mathcal{D}) \, \mathrm{d}\tilde{\boldsymbol{w}} \, + \frac{1}{2} \, + \, \log(2\pi) \, .$$

## B.2  Mixture Importance Sampling for Variance Reduction

The epistemic uncertainties in Eq. (1) and Eq. (2) are expectations of KL divergences over the posterior. We have to approximate these integrals.

If the posterior has different modes, a concentrated importance sampling function has a high variance of estimates, therefore converges very slowly [Steele et al., 2006]. Thus, we use mixture importance sampling (MIS) [Hesterberg, 1995]. MIS uses a mixture model for sampling, instead of a unimodal model of standard importance sampling [Owen and Zhou, 2000]. Multiple importance sampling Veach and Guibas [1995] is similar to MIS and equal to it for balanced heuristics [Owen and Zhou, 2000]. More details on these and similar methods can be found in Owen and Zhou [2000], Cappé et al. [2004], Elvira et al. [2015, 2019], Steele et al. [2006], Raftery and Bao [2010]. MIS has been very successfully applied to estimate multimodal densities. For example, the evidence lower bound (ELBO) [Kingma and Welling, 2014] has been improved by multiple importance sampling ELBO [Kviman et al., 2022]. Using a mixture model should ensure that at least one of its components will locally match the shape of the integrand. Often, MIS iteratively enrich the sampling distribution by new modes [Raftery and Bao, 2010].

In contrast to iterative enrichment, which finds modes by chance, we are able to explicitly search for posterior modes, where the integrand of the definition of epistemic uncertainty is large. For each of these modes, we define a component of the mixture from which we then sample. We have the huge advantage to have explicit expressions for the integrand. The integrand of the epistemic uncertainty in Eq. (1) and Eq. (2) has the form

$$\mathrm{D}(p(\boldsymbol{y} \mid \boldsymbol{x}, \boldsymbol{w}), \, p(\boldsymbol{y} \mid \boldsymbol{x}, \tilde{\boldsymbol{w}})) \, p(\tilde{\boldsymbol{w}} \mid \mathcal{D}) \, , \tag{30}$$

where $\mathrm{D}(\cdot\,,\,\cdot)$ is a distance or divergence of distributions which is computed using the parameters that determine those distributions. The distance/divergence $\mathrm{D}(\cdot\,,\,\cdot)$ eliminates the aleatoric uncertainty, which is present in $p(\boldsymbol{y}\mid\boldsymbol{x},\boldsymbol{w})$ and $p(\boldsymbol{y}\mid\boldsymbol{x},\tilde{\boldsymbol{w}})$. Essentially, $\mathrm{D}(\cdot\,,\,\cdot)$ reduces distributions to functions of their parameters.

Importance sampling is applied to estimate integrals of the form

$$s = \int_{\mathcal{X}} f(\boldsymbol{x})\, p(\boldsymbol{x})\, \mathrm{d}\boldsymbol{x} = \int_{\mathcal{X}} \frac{f(\boldsymbol{x})\, p(\boldsymbol{x})}{q(\boldsymbol{x})}\, q(\boldsymbol{x})\, \mathrm{d}\boldsymbol{x}\,, \tag{31}$$

with integrand $f(x)$ and probability distributions $p(\boldsymbol{x})$ and $q(\boldsymbol{x})$, when it is easier to sample according to $q(\boldsymbol{x})$ than $p(\boldsymbol{x})$. The estimator of Eq. (31) when drawing $\boldsymbol{x}_n$ according to $q(\boldsymbol{x})$ is given by

$$\hat{s} = \frac{1}{N} \sum_{n=1}^{N} \frac{f(\boldsymbol{x}_n)\, p(\boldsymbol{x}_n)}{q(\boldsymbol{x}_n)}\,. \tag{32}$$

The asymptotic variance $\sigma_s^2$ of importance sampling is given by (see e.g. Owen and Zhou [2000]):

$$\sigma_s^2 = \int_{\mathcal{X}} \left( \frac{f(\boldsymbol{x})\, p(\boldsymbol{x})}{q(\boldsymbol{x})} - s \right)^2 q(\boldsymbol{x})\, \mathrm{d}\boldsymbol{x} \tag{33}$$

$$= \int_{\mathcal{X}} \left( \frac{f(\boldsymbol{x})\, p(\boldsymbol{x})}{q(\boldsymbol{x})} \right)^2 q(\boldsymbol{x})\, \mathrm{d}\boldsymbol{x} - s^2\,,$$

and its estimator when drawing $\boldsymbol{x}_n$ from $q(\boldsymbol{x})$ is given by

$$\hat{\sigma}_s^2 = \frac{1}{N} \sum_{n=1}^{N} \left( \frac{f(\boldsymbol{x}_n)\, p(\boldsymbol{x}_n)}{q(\boldsymbol{x}_n)} - s \right)^2 \tag{34}$$

$$= \frac{1}{N} \sum_{n=1}^{N} \left( \frac{f(\boldsymbol{x}_n)\, p(\boldsymbol{x}_n)}{q(\boldsymbol{x}_n)} \right)^2 - s^2\,.$$

We observe, that the variance is determined by the term $\frac{f(\boldsymbol{x})p(\boldsymbol{x})}{q(\boldsymbol{x})}$, thus we want $q(\boldsymbol{x})$ to be proportional to $f(\boldsymbol{x})p(\boldsymbol{x})$. Most importantly, $q(\boldsymbol{x})$ should not be close to zero for large $f(\boldsymbol{x})p(\boldsymbol{x})$. To give an intuition about the severity of unmatched modes, we depict an educational example in Fig. B.1. Now we plug in the form of the integrand given by Eq. (30) into Eq. (31), to calculate the expected divergence $\mathrm{D}(\cdot\,,\,\cdot)$ under the model posterior $p(\tilde{\boldsymbol{w}}\mid\mathcal{D})$. This results in

$$v = \int_{\mathcal{W}} \frac{\mathrm{D}(p(\boldsymbol{y}\mid\boldsymbol{x},\boldsymbol{w})\,,\, p(\boldsymbol{y}\mid\boldsymbol{x},\tilde{\boldsymbol{w}}))\, p(\tilde{\boldsymbol{w}}\mid\mathcal{D})}{q(\tilde{\boldsymbol{w}})}\, q(\tilde{\boldsymbol{w}})\, \mathrm{d}\tilde{\boldsymbol{w}}\,, \tag{35}$$

with estimate

$$\hat{v} = \frac{1}{N} \sum_{n=1}^{N} \frac{\mathrm{D}(p(\boldsymbol{y}\mid\boldsymbol{x},\boldsymbol{w})\,,\, p(\boldsymbol{y}\mid\boldsymbol{x},\tilde{\boldsymbol{w}}_n))\, p(\tilde{\boldsymbol{w}}_n\mid\mathcal{D})}{q(\tilde{\boldsymbol{w}}_n)}\,. \tag{36}$$

The variance is given by

$$\sigma_v^2 = \int_{\mathcal{W}} \left( \frac{\mathrm{D}(p(\boldsymbol{y}\mid\boldsymbol{x},\boldsymbol{w})\,,\, p(\boldsymbol{y}\mid\boldsymbol{x},\tilde{\boldsymbol{w}}))\, p(\tilde{\boldsymbol{w}}\mid\mathcal{D})}{q(\tilde{\boldsymbol{w}})} - v \right)^2 q(\tilde{\boldsymbol{w}})\, \mathrm{d}\tilde{\boldsymbol{w}} \tag{37}$$

$$= \int_{\mathcal{W}} \left( \frac{\mathrm{D}(p(\boldsymbol{y}\mid\boldsymbol{x},\boldsymbol{w})\,,\, p(\boldsymbol{y}\mid\boldsymbol{x},\tilde{\boldsymbol{w}}))\, p(\tilde{\boldsymbol{w}}\mid\mathcal{D})}{q(\tilde{\boldsymbol{w}})} \right)^2 q(\tilde{\boldsymbol{w}})\, \mathrm{d}\tilde{\boldsymbol{w}} - v^2\,.$$

The estimate for the variance is given by

$$\hat{\sigma}_v^2 = \frac{1}{N} \sum_{n=1}^{N} \left( \frac{\mathrm{D}(p(\boldsymbol{y}\mid\boldsymbol{x},\boldsymbol{w})\,,\, p(\boldsymbol{y}\mid\boldsymbol{x},\tilde{\boldsymbol{w}}_n))\, p(\tilde{\boldsymbol{w}}_n\mid\mathcal{D})}{q(\tilde{\boldsymbol{w}}_n)} - v \right)^2 \tag{38}$$

$$= \frac{1}{N} \sum_{n=1}^{N} \left( \frac{\mathrm{D}(p(\boldsymbol{y}\mid\boldsymbol{x},\boldsymbol{w})\,,\, p(\boldsymbol{y}\mid\boldsymbol{x},\tilde{\boldsymbol{w}}_n))\, p(\tilde{\boldsymbol{w}}_n\mid\mathcal{D})}{q(\tilde{\boldsymbol{w}}_n)} \right)^2 - v^2\,,$$

where $\tilde{\boldsymbol{w}}_n$ is drawn according to $q(\tilde{\boldsymbol{w}})$. The asymptotic ($N \to \infty$) confidence intervals are given by

$$\lim_{N \to \infty} \Pr\left(-a\,\frac{\sigma_v}{\sqrt{N}} \;\leqslant\; \hat{v} \;-\; v \;\leqslant\; b\,\frac{\sigma_v}{\sqrt{N}}\right) \;=\; \frac{1}{\sqrt{2\,\pi}} \int_{-a}^{b} \exp(-\,1/2\,t^2)\,\mathrm{d}t\,. \qquad (39)$$

Thus, $\hat{v}$ converges with $\frac{\sigma_v}{\sqrt{N}}$ to $v$. The asymptotic confidence interval is proofed in Weinzierl [2000] and Hesterberg [1996] using the Lindeberg–Lévy central limit theorem which ensures the asymptotic normality of the estimate $\hat{v}$. The $q(\tilde{\boldsymbol{w}})$ that minimizes the variance is

$$q(\tilde{\boldsymbol{w}}) = \frac{\mathrm{D}(p(\boldsymbol{y} \mid \boldsymbol{x}, \boldsymbol{w})\,,\,p(\boldsymbol{y} \mid \boldsymbol{x}, \tilde{\boldsymbol{w}}))\,p(\tilde{\boldsymbol{w}} \mid \mathcal{D})}{v}\,. \qquad (40)$$

Thus we want to find a density $q(\tilde{\boldsymbol{w}})$ that is proportional to $\mathrm{D}(p(\boldsymbol{y} \mid \boldsymbol{x}, \boldsymbol{w})\,,\,p(\boldsymbol{y} \mid \boldsymbol{x}, \tilde{\boldsymbol{w}}))\,p(\tilde{\boldsymbol{w}} \mid \mathcal{D})$. Only approximating the posterior $p(\tilde{\boldsymbol{w}} \mid \mathcal{D})$ as Deep Ensembles or MC dropout is insufficient to guarantee a low expected error, since the sampling variance cannot be bounded, as $\sigma_v^2$ could get arbitrarily big if the distance is large but the probability under the sampling distribution is very small. For $q(\tilde{\boldsymbol{w}}) \propto p(\tilde{\boldsymbol{w}} \mid \mathcal{D})$ and non-negative, unbounded, but continuous $\mathrm{D}(\cdot\,,\,\cdot)$, the variance $\sigma_v^2$ given by Eq. (37) cannot be bounded.

For example, if $\mathrm{D}(\cdot\,,\,\cdot)$ is the KL-divergence and both $p(\boldsymbol{y} \mid \boldsymbol{x}, \boldsymbol{w})$ and $p(\boldsymbol{y} \mid \boldsymbol{x}, \tilde{\boldsymbol{w}})$ are Gaussians where the means $\mu(\boldsymbol{x}, \boldsymbol{w})$, $\mu(\boldsymbol{x}, \tilde{\boldsymbol{w}})$ and variances $\sigma^2(\boldsymbol{x}, \boldsymbol{w})$, $\sigma^2(\boldsymbol{x}, \tilde{\boldsymbol{w}})$ are estimates provided by the models, the KL is unbounded. The KL divergence between two Gaussian distributions is given by

$$\mathrm{D}_{\mathrm{KL}}(p(y \mid \boldsymbol{x}, \boldsymbol{w}) \,\|\, p(y \mid \boldsymbol{x}, \tilde{\boldsymbol{w}})) \qquad (41)$$
$$= -\int_{\mathcal{Y}} p(y \mid \boldsymbol{x}, \boldsymbol{w}) \log\left(\frac{p(y \mid \boldsymbol{x}, \boldsymbol{w})}{p(y \mid \boldsymbol{x}, \tilde{\boldsymbol{w}})}\right)\,\mathrm{d}y$$
$$= \frac{1}{2} \log\left(\frac{\sigma^2(\boldsymbol{x}, \tilde{\boldsymbol{w}})}{\sigma^2(\boldsymbol{x}, \boldsymbol{w})}\right) + \frac{\sigma^2(\boldsymbol{x}, \boldsymbol{w}) + (\mu(\boldsymbol{x}, \boldsymbol{w}) - \mu(\boldsymbol{x}, \tilde{\boldsymbol{w}}))^2}{2\,\sigma^2(\boldsymbol{x}, \tilde{\boldsymbol{w}})} - \frac{1}{2}\,.$$

For $\sigma^2(\boldsymbol{x}, \tilde{\boldsymbol{w}})$ going towards zero and a non-zero difference of the mean values, the KL-divergence can be arbitrarily large. Therefore, methods that only consider the posterior $p(\tilde{\boldsymbol{w}} \mid \mathcal{D})$ cannot bound the variance $\sigma_v^2$ if $\mathrm{D}(\cdot\,,\,\cdot)$ is unbounded and the parameters $\tilde{\boldsymbol{w}}$ allow distributions which can make $\mathrm{D}(\cdot\,,\,\cdot)$ arbitrary large.

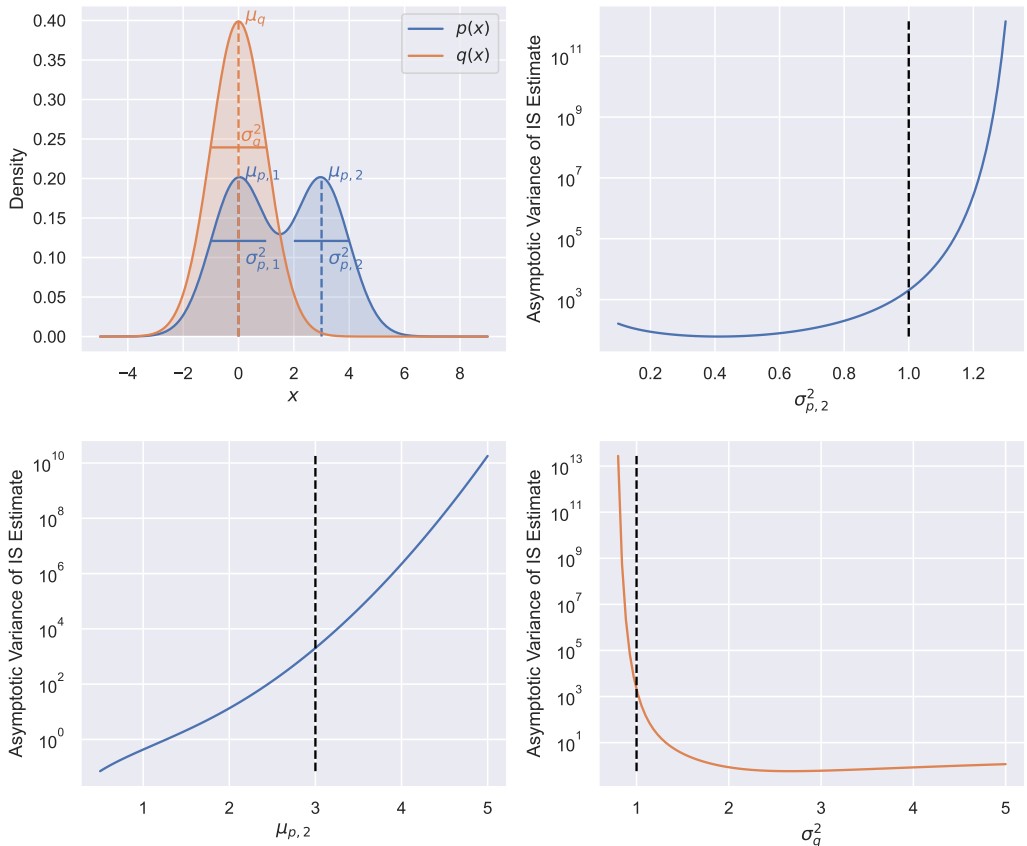

Figure B.1: Analysis of asymptotic variance of importance sampling for multimodal target distribution $p(x)$ and a unimodal sampling distribution $q(x)$. The target distribution is a mixture of two Gaussian distributions with means $\mu_{p,1}, \mu_{p,2}$ and variances $\sigma_{p,1}^2, \sigma_{p,2}^2$. The sampling distribution is a single Gaussian with mean $\mu_q$ and variance $\sigma_q^2$. $q(x)$ matches one of the modes of $p(x)$, but misses the other. Both distributions are visualized for their standard parameters $\mu_{p,1} = \mu_q = 0$, $\mu_{p,2} = 3$ and $\sigma_{p,1}^2 = \sigma_{p,2}^2 = \sigma_q^2 = 1$, where both mixture components of $p(x)$ are equally weighted. We calculate the asymptotic variance (Eq. (33) with $f(\boldsymbol{x}) = 1$) for different values of $\sigma_{p,2}^2$, $\mu_{p,2}$ and $\sigma_q^2$ and show the results in the top right, bottom left and bottom right plot respectively. The standard value for the varied parameter is indicated by the black dashed line. We observe, that slightly increasing the variance of the second mixture component of $p(x)$, which is not matched by the mode of $q(x)$, rapidly increases the asymptotic variance. Similarly, increasing the distance between the center of the unmatched mixture component of $p(x)$ and $q(x)$ strongly increases the asymptotic variance. On the contrary, increasing the variance of the sampling distribution $q(x)$ does not lead to a strong increase, as the worse approximation of the matched mode of $p(x)$ is counterbalanced by putting probability mass where the second mode of $p(x)$ is located. Note, that this issue is even more exacerbated if $f(x)$ is non-constant. Then, $q(x)$ has to match the modes of $f(x)$ as well.

# C  Experimental Details and Further Experiments

Our code is publicly available at https://github.com/ml-jku/quam.

## C.1  Details on the Adversarial Model Search

During the adversarial model search, we seek to maximize the KL divergence between the prediction of the reference model and adversarial models. For an example, see Fig. C.1. We found that directly maximizing the KL divergence always leads to similar solutions to the optimization problem. Therefore, we maximized the likelihood of a new test point to be in each possible class. The optimization problem is very similar, considering the predictive distribution $p(\boldsymbol{y} \mid \boldsymbol{x}, \boldsymbol{w})$ of a reference model and the predictive distribution $p(\boldsymbol{y} \mid \boldsymbol{x}, \tilde{\boldsymbol{w}})$ of an adversarial model, the model that is updated. The KL divergence between those two is given by

$$
\mathrm{D_{KL}}(p(\boldsymbol{y} \mid \boldsymbol{x}, \boldsymbol{w}) \,\|\, p(\boldsymbol{y} \mid \boldsymbol{x}, \tilde{\boldsymbol{w}})) \tag{42}
$$
$$
= \sum p(\boldsymbol{y} \mid \boldsymbol{x}, \boldsymbol{w}) \log\left(\frac{p(\boldsymbol{y} \mid \boldsymbol{x}, \boldsymbol{w})}{p(\boldsymbol{y} \mid \boldsymbol{x}, \tilde{\boldsymbol{w}})}\right)
$$
$$
= \sum p(\boldsymbol{y} \mid \boldsymbol{x}, \boldsymbol{w}) \log\left(p(\boldsymbol{y} \mid \boldsymbol{x}, \boldsymbol{w})\right) - \sum p(\boldsymbol{y} \mid \boldsymbol{x}, \boldsymbol{w}) \log\left(p(\boldsymbol{y} \mid \boldsymbol{x}, \tilde{\boldsymbol{w}})\right)
$$
$$
= -\mathrm{H}[p(\boldsymbol{y} \mid \boldsymbol{x}, \boldsymbol{w})] + \mathrm{CE}[p(\boldsymbol{y} \mid \boldsymbol{x}, \boldsymbol{w}),\, p(\boldsymbol{y} \mid \boldsymbol{x}, \tilde{\boldsymbol{w}})] \,.
$$

Only the cross-entropy between the predictive distributions of the reference model parameterized by $\boldsymbol{w}$ and the adversarial model parameterized by $\tilde{\boldsymbol{w}}$ plays a role in the optimization, since the entropy of $p_{\boldsymbol{w}}$ stays constant during the adversarial model search. Thus, the optimization target is equivalent to the cross-entropy loss, except that $\boldsymbol{p_w}$ is generally not one-hot encoded but an arbitrary categorical distribution. This also relates to targeted / untargeted adversarial attacks on the input. Targeted attacks try to maximize the output probability of a specific class. Untargeted attacks try to minimize the probability of the originally predicted class, by maximizing all other classes. We found that attacking individual classes works better empirically, while directly maximizing the KL divergence always leads to similar solutions for different searches. The result often is a further increase of the probability associated with the most likely class. Therefore, we conducted as many adversarial model searches for a new test point, as there are classes in the classification task. Thereby, we optimize the cross-entropy loss for one specific class in each search.

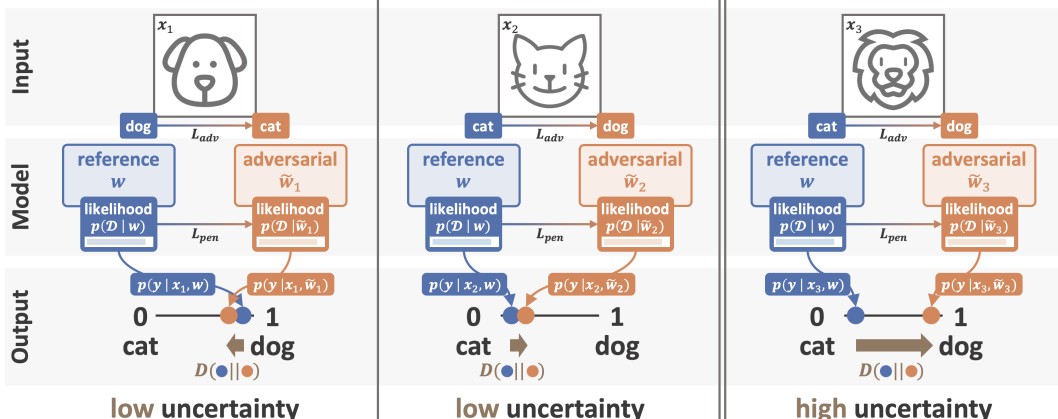

Figure C.1: Illustrative example of QUAM. We illustrate quantifying the predictive uncertainty of a given, pre-selected model (blue), a classifier for images of cats and dogs. For each of the input images, we search for adversarial models (orange) that make different predictions than the given, pre-selected model while explaining the training data equally well (having a high likelihood). The adversarial models found for an image of a dog or a cat still make similar predictions (low epistemic uncertainty), while the adversarial model found for an image of a lion makes a highly different prediction (high epistemic uncertainty), as features present in images of both cats and dogs can be utilized to classify the image of a lion.

For regression, we add a small perturbation to the bias of the output linear layer. This is necessary to ensure a gradient in the first update step, as the model to optimize is initialized with the reference model. For regression, we perform the adversarial model search two times, as the output of an adversarial model could be higher or lower than the reference model if we assume a scalar output. We force, that the two adversarial model searches get higher or lower outputs than the reference model respectively. While the loss of the reference model on the training dataset $L_{ref}$ is calculated on the full training dataset (as it has to be done only once), we approximate $L_{pen}$ by randomly drawn mini-batches for each update step. Therefore, the boundary condition might not be satisfied on the full training set, even if the boundary condition is satisfied for the mini-batch estimate.

As described in the main paper, the resulting model of each adversarial model search is used to define the location of a mixture component of a sampling distribution $q(\tilde{\boldsymbol{w}})$ (Eq. (6)). The epistemic uncertainty is estimated by Eq. (4), using models sampled from this mixture distribution. The simplest choice of distributions for each mixture distribution is a delta distribution at the location of the adversarial model $\breve{\boldsymbol{w}}_k$. While this performs well empirically, we discard a lot of information by not utilizing predictions of models obtained throughout the adversarial model search. The intermediate solutions of the adversarial model search allow to assess how easily models with highly divergent predictive distributions to the reference model can be found. Furthermore, the expected mean squared error (Eq. (5)) decreases with $\frac{1}{N}$ with the number of samples $N$ and the expected variance of the estimator (Eq. (38)) decreases with $\frac{1}{\sqrt{N}}$. Therefore, using more samples is beneficial empirically, even though we potentially introduce a bias to the estimator.

Consequently, we utilize all sampled models during the adversarial model search as an empirical sampling distribution for our experiments. This is the same as how members of an ensemble can be seen as an empirical sampling distribution [Gustafsson et al., 2020] and conceptually similar to Snapshot ensembling [Huang et al., 2017]. To compute Eq. (4), we use the negative exponential training loss of each model to approximate its posterior probability $p(\tilde{\boldsymbol{w}} \mid \mathcal{D})$. Note that the training loss is the negative log-likelihood, which in turn is proportional to the posterior probability. Note we temperature-scale the approximate posterior probability by $p(\tilde{\boldsymbol{w}} \mid \mathcal{D})^{\frac{1}{T}}$, with the temperature parameter $T$ set as a hyperparameter.

## C.2 Simplex Example

We sample the training dataset $\mathcal{D} = \{(\boldsymbol{x}_k, \boldsymbol{y}_k)\}_{k=1}^K$ from three Gaussian distributions (21 datapoints from each Gaussian) at locations $\boldsymbol{\mu}_1 = (-4, -2)^T$, $\boldsymbol{\mu}_2 = (4, -2)^T$, $\boldsymbol{\mu}_3 = (0, 2\sqrt{2})^T$ and the same two-dimensional covariance with $\sigma^2 = 1.5$ on both entries of the diagonal and zero on the off-diagonals. The labels $\boldsymbol{y}_k$ are one-hot encoded vectors, signifying which Gaussian the input $\boldsymbol{x}_k$ was sampled from. The new test point $\boldsymbol{x}$ we evaluate for is located at $(-6, 2)$. To attain the likelihood

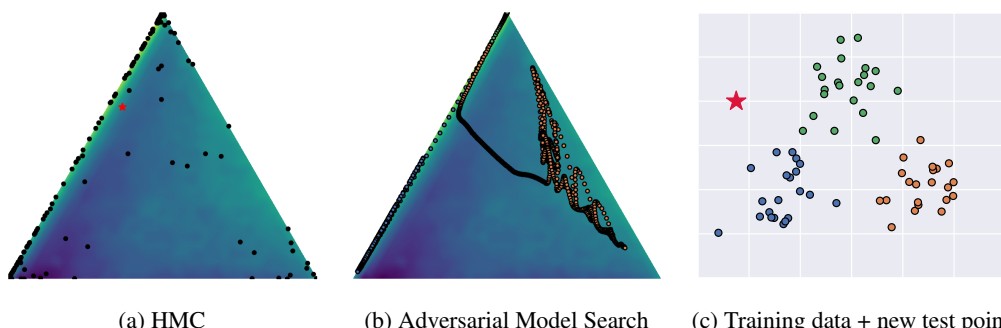

|            (a) HMC            |   (b) Adversarial Model Search   |   (c) Training data + new test point   |

Figure C.2: Softmax outputs (black) of individual models of HMC (a) as well as their average output (red) on a probability simplex. Softmax outputs of models found throughout the adversarial model search (b), colored by the attacked class. Left, right and top corners denote 100% probability mass at the blue, orange and green class in (c) respectively. Models were selected on the training data, and evaluated on the new test point (red) depicted in (c). The background color denotes the maximum likelihood of the training data that is achievable by a model having equal softmax output as the respective location on the simplex.

for each position on the probability simplex, we train a two-layer fully connected neural network (with parameters $\boldsymbol{w}$) with hidden size of 10 on this dataset. We minimize the combined loss

$$\mathrm{L} = \frac{1}{K} \sum_{k=1}^{K} l(p(\boldsymbol{y} \mid \boldsymbol{x}_k, \boldsymbol{w}), \boldsymbol{y}_k) + l(p(\boldsymbol{y} \mid \boldsymbol{x}, \boldsymbol{w}), \breve{\boldsymbol{y}}) , \qquad (43)$$

where $l$ is the cross-entropy loss function and $\breve{\boldsymbol{y}}$ is the desired categorical distribution for the output of the network. We report the likelihood on the training dataset upon convergence of the training procedure for $\breve{\boldsymbol{y}}$ on the probability simplex. To average over different initializations of $\boldsymbol{w}$ and alleviate the influence of potentially bad local minima, we use the median over 20 independent runs to calculate the maximum.

For all methods, we utilize the same two-layer fully connected neural network with hidden size of 10; for MC dropout we additionally added dropout with dropout probability 0.2 after every intermediate layer. We trained 50 networks for the Deep Ensemble results. For MC dropout we sampled predictive distributions using 1000 forward passes.

Fig. C.2 (a) shows models sampled using HMC, which is widely regarded as the best approximation to the ground truth for predictive uncertainty estimation. Furthermore, Fig. C.2 (b) shows models obtained by executing the adversarial model search for the given training dataset and test point depicted in Fig. C.2 (c). HMC also provides models that put more probability mass on the orange class. Those are missed by Deep Ensembles and MC dropout (see Fig. 2 (a) and (b)). The adversarial model search used by QUAM helps to identify those regions.

### C.3 Epistemic Uncertainty on Synthetic Dataset

We create the two-moons dataset using the implementation of Pedregosa et al. [2011]. All experiments were performed on a three-layer fully connected neural network with hidden size 100 and ReLU activations. For MC dropout, dropout with dropout probability of 0.2 was applied after the intermediate layers. We assume to have a trained reference model $\boldsymbol{w}$ of this architecture. Results of the same runs as in the main paper, but calculated for the epistemic uncertainty in setting (b) (see Eq. (2)) are depicted in Fig. C.3. Again, QUAM matches the ground truth best.

Furthermore, we conducted experiments on a synthetic regression dataset, where the input feature $x$ is drawn randomly between $[-\pi, \pi]$ and the target is $y = \sin(x) + \epsilon$, with $\epsilon \sim \mathcal{N}(0, 0.1)$. The results are depicted in Fig. C.4. As for the classification results, the estimate of QUAM is closest to the ground truth provided by HMC.

The HMC implementation of Cobb and Jalaian [2021] was used to obtain the ground truth epistemic uncertainties. For the Laplace approximation, we used the implementation of Daxberger et al. [2021]. For SG-MCMC we used the python package of Kapoor [2023].

### C.4 Epistemic Uncertainty on Vision Datasets

Several vision datasets and their corresponding OOD datasets are commonly used for benchmarking predictive uncertainty quantification in the literature, e.g. in Blundell et al. [2015], Gal and Ghahramani [2016], Malinin and Gales [2018], Ovadia et al. [2019], van Amersfoort et al. [2020], Mukhoti et al. [2021], Postels et al. [2021], Band et al. [2022]. Our experiments focused on two of those: MNIST [LeCun et al., 1998] and its OOD derivatives as the most basic benchmark and ImageNet1K [Deng et al., 2009] to demonstrate our method's ability to perform on a larger scale. Four types of experiments were performed: (i) OOD detection (ii) adversarial example detection, (iii) misclassification detection and (iv) selective prediction. Our experiments on adversarial example detection did not utilize a specific adversarial attack on the input images, but natural adversarial examples [Hendrycks et al., 2021], which are images from the ID classes, but wrongly classified by standard ImageNet classifiers. Misclassification detection and selective prediction was only performed for Imagenet1K, since MNIST classifiers easily reach accuracies of 99% on the test set, thus hardly misclassifying any samples. In all cases except selective prediciton, we measured AUROC, FPR at TPR of 95% and AUPR of classifying ID vs. OOD, non-adversarial vs. adversarial and correctly classified vs. misclassified samples (on ID test set), using the epistemic uncertainty estimate provided by the different methods. For selective prediction, we utilized the epistemic uncertainty estimate to select a subset of samples on the ID test set.

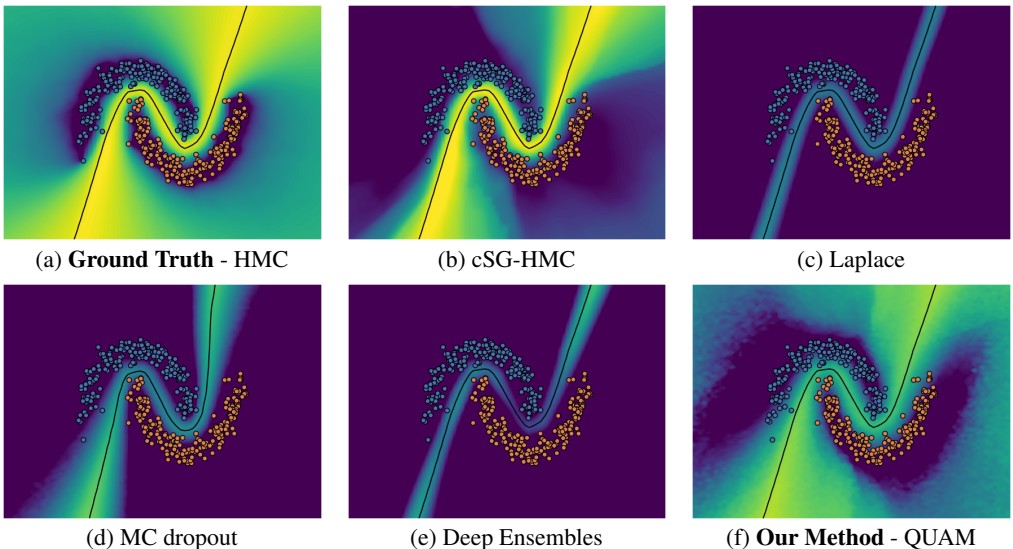

Figure C.3: Epistemic uncertainty as in Eq. (2). Yellow denotes high epistemic uncertainty. Purple denotes low epistemic uncertainty. The black lines show the decision boundary of the reference model $w$. HMC is considered to be the ground truth epistemic uncertainty. The estimate of QUAM is closest to the ground truth. All other methods underestimate the epistemic uncertainty in the top left and bottom right corner, as all models sampled by those predict the same class with high confidence for those regions.

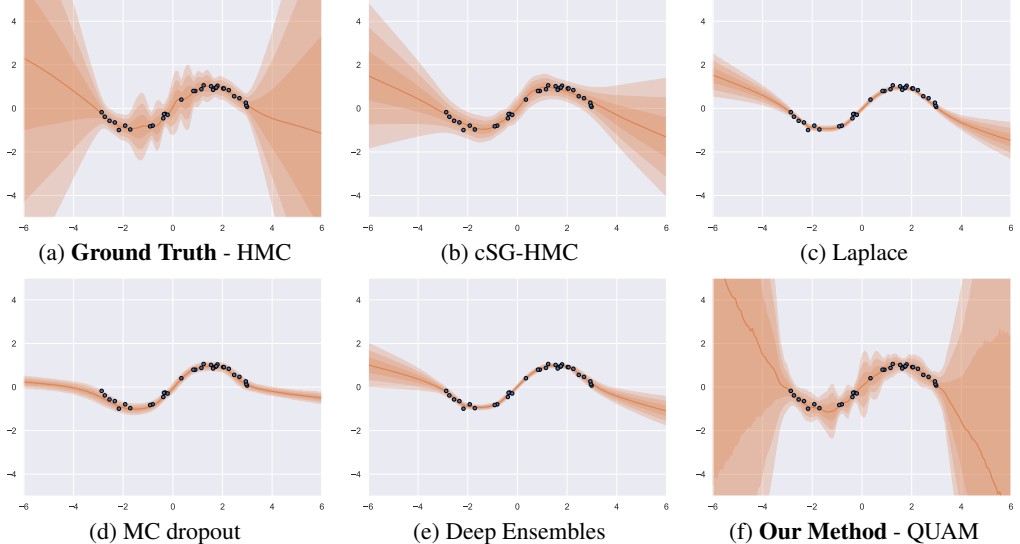

Figure C.4: Variance between different models found by different methods on synthetic `sine` dataset. Orange line denotes the empirical mean of the averaged models, shades denote one, two and three standard deviations respectively. HMC is considered to be the ground truth epistemic uncertainty. The estimate of QUAM is closest to the ground truth. All other methods fail to capture the variance between points as well as the variance left outside the region ($[-\pi, \pi]$) datapoints are sampled from.

Table C.1: Additional baseline MoLA: AUROC using the epistemic uncertainty of a given, pre-selected model as a score to distinguish between ID (MNIST) and OOD samples. Results for additional baseline method MoLA, comparing to Laplace approximation, Deep Ensembles (DE) and QUAM. Results are averaged over three independent runs.

| $\mathcal{D}_{\text{ood}}$ | Laplace | MoLA | DE | QUAM |
|---|---|---|---|---|
| FMNIST | $.978_{\pm.004}$ | $.986_{\pm.002}$ | $.988_{\pm.001}$ | $\mathbf{.994}_{\pm.001}$ |
| KMNIST | $.959_{\pm.006}$ | $.984_{\pm.000}$ | $.990_{\pm.001}$ | $\mathbf{.994}_{\pm.001}$ |
| EMNIST | $.877_{\pm.011}$ | $.920_{\pm.002}$ | $.924_{\pm.003}$ | $\mathbf{.937}_{\pm.008}$ |
| OMNIGLOT | $.963_{\pm.003}$ | $.979_{\pm.000}$ | $.983_{\pm.001}$ | $\mathbf{.992}_{\pm.001}$ |

### C.4.1 MNIST

OOD detection experiments were performed on MNIST with FashionMNIST (FMNIST) [Xiao et al., 2017], EMNIST [Cohen et al., 2017], KMNIST [Clanuwat et al., 2018] and OMNIGLOT [Lake et al., 2015] as OOD datasets. In case of EMNIST, we only used the "letters" subset, thus excluding classes overlapping with MNIST (digits). We used the MNIST (test set) vs FMNIST (train set) OOD detection task to tune hyperparameters for all methods. The evaluation was performed using the complete test sets of the above-mentioned datasets ($n = 10000$).

For each seed, a separate set of Deep Ensembles was trained. Ensembles with the size of 10 were found to perform best. MC dropout was used with a number of samples set to 2048. This hyperparameter setting was found to perform well. A higher sampling size would increase the performance marginally while increasing the computational load. Noteworthy is the fact, that with these settings the computational requirements of MC dropout surpassed those of QUAM. Laplace approximation was performed only for the last layer, due to the computational demand making it infeasible on the full network with our computational capacities. Mixture of Laplace approximations Eschenhagen et al. [2021] was evaluated as well using the parameters provided in the original work. Notably, the results from the original work suggesting improved performance compared to the Deep Ensembles on these tasks could not be reproduced. Comparison is provided in Table C.1. SG-HMC was performed on the full network using the Python package from Kapoor [2023]. Parameters were set in accordance with those of the original authors [Zhang et al., 2020]. For QUAM, the initial penalty parameter found by tuning was $c_0 = 6$, which was exponentially increased ($c_{t+1} = \eta c_t$) with $\eta = 2$ every 14 gradient steps for a total of two epochs through the training dataset. Gradient steps were performed using Adam [Kingma and Ba, 2014] with a learning rate of $5.e\text{-}3$ and weight decay of $1.e\text{-}3$, chosen equivalent to the original training parameters of the model. A temperature of $1.e\text{-}3$ was used for scaling the cross-entropy loss, an approximation for the posterior probabilities when calculating Eq. (4). Detailed results and additional metrics and replicates of the experiments can be found in Tab. C.2. Experiments were performed three times with seeds: {42, 142, 242} to provide confidence intervals. Histograms of the scores on the ID dataset and the OOD datasets for different methods are depicted in Fig. C.5.

### C.4.2 ImageNet

For ImageNet1K [Deng et al., 2009], OOD detection experiments were performed with ImageNet-O [Hendrycks et al., 2021], adversarial example detection experiments with ImageNet-A [Hendrycks et al., 2021], and misclassification detection as well as selective prediction experiments on the official validation set of ImageNet1K. For each experiment, we utilized a pre-trained EfficientNet [Tan and Le, 2019] architecture with 21.5 million trainable weights available through PyTorch [Paszke et al., 2019], achieving a top-1 accuracy of $84.2\%$ as well as a top-5 accuracy of $96.9\%$.

cSG-HMC was performed on the last layer using the best hyperparameters that resulted from a hyperparameter search around the ones suggested by the original authors [Zhang et al., 2020]. The Laplace approximation with the implementation of [Daxberger et al., 2021] was not feasible to compute for this problem on our hardware, even only for the last layer. Similarly to the experiments in section C.4.1, we compare against a Deep Ensemble consisting of 10 pre-trained EfficientNet architectures ranging from 5.3 million to 66.3 million trainable weights (DE (all)). Also, we retrained the last layer of 10 ensemble members (DE (LL)) given the same base network. We also compare

Table C.2: Detailed results of MNIST OOD detection experiments, reporting AUROC, AUPR and FPR@TPR=95% for individual seeds.

| OOD dataset | Method | Seed | ↑ AUPR | ↑ AUROC | ↓ FPR@TPR=95% |
|---|---|---|---|---|---|
| EMNIST | cSG-HMC | 42 | 0.8859 | 0.8823 | 0.5449 |
| | | 142 | 0.8714 | 0.8568 | 0.8543 |
| | | 242 | 0.8797 | 0.8673 | 0.7293 |
| | Laplace | 42 | 0.8901 | 0.8861 | 0.5273 |
| | | 142 | 0.8762 | 0.8642 | 0.7062 |
| | | 242 | 0.8903 | 0.8794 | 0.6812 |
| | Deep Ensembles | 42 | 0.9344 | 0.9239 | 0.4604 |
| | | 142 | 0.9325 | 0.9236 | 0.4581 |
| | | 242 | 0.9354 | 0.9267 | 0.4239 |
| | MC dropout | 42 | 0.8854 | 0.8787 | 0.5636 |
| | | 142 | 0.8769 | 0.8630 | 0.6718 |
| | | 242 | 0.8881 | 0.8751 | 0.6855 |
| | QUAM | 42 | 0.9519 | 0.9454 | 0.3405 |
| | | 142 | 0.9449 | 0.9327 | 0.4538 |
| | | 242 | 0.9437 | 0.9317 | 0.4325 |
| FMNIST | cSG-HMC | 42 | 0.9532 | 0.9759 | 0.0654 |
| | | 142 | 0.9610 | 0.9731 | 0.0893 |
| | | 242 | 0.9635 | 0.9827 | 0.0463 |
| | Laplace | 42 | 0.9524 | 0.9754 | 0.0679 |
| | | 142 | 0.9565 | 0.9739 | 0.0788 |
| | | 242 | 0.9613 | 0.9824 | 0.0410 |
| | Deep Ensembles | 42 | 0.9846 | 0.9894 | 0.0319 |
| | | 142 | 0.9776 | 0.9865 | 0.0325 |
| | | 242 | 0.9815 | 0.9881 | 0.0338 |
| | MC dropout | 42 | 0.9595 | 0.9776 | 0.0644 |
| | | 142 | 0.9641 | 0.9748 | 0.0809 |
| | | 242 | 0.9696 | 0.9848 | 0.0393 |
| | QUAM | 42 | 0.9896 | 0.9932 | 0.0188 |
| | | 142 | 0.9909 | 0.9937 | 0.0210 |
| | | 242 | 0.9925 | 0.9952 | 0.0132 |
| KMNIST | cSG-HMC | 42 | 0.9412 | 0.9501 | 0.2092 |
| | | 142 | 0.9489 | 0.9591 | 0.1551 |
| | | 242 | 0.9505 | 0.9613 | 0.1390 |
| | Laplace | 42 | 0.9420 | 0.9520 | 0.1915 |
| | | 142 | 0.9485 | 0.9617 | 0.1378 |
| | | 242 | 0.9526 | 0.9640 | 0.1165 |
| | Deep Ensembles | 42 | 0.9885 | 0.9899 | 0.0417 |
| | | 142 | 0.9875 | 0.9891 | 0.0458 |
| | | 242 | 0.9884 | 0.9896 | 0.0473 |
| | MC dropout | 42 | 0.9424 | 0.9506 | 0.2109 |
| | | 142 | 0.9531 | 0.9618 | 0.1494 |
| | | 242 | 0.9565 | 0.9651 | 0.1293 |
| | QUAM | 42 | 0.9928 | 0.9932 | 0.0250 |
| | | 142 | 0.9945 | 0.9952 | 0.0194 |
| | | 242 | 0.9925 | 0.9932 | 0.0260 |
| OMNIGLOT | cSG-HMC | 42 | 0.9499 | 0.9658 | 0.1242 |
| | | 142 | 0.9459 | 0.9591 | 0.1498 |
| | | 242 | 0.9511 | 0.9637 | 0.1222 |
| | Laplace | 42 | 0.9485 | 0.9647 | 0.1238 |
| | | 142 | 0.9451 | 0.9597 | 0.1345 |
| | | 242 | 0.9526 | 0.9656 | 0.1077 |
| | Deep Ensembles | 42 | 0.9771 | 0.9822 | 0.0621 |
| | | 142 | 0.9765 | 0.9821 | 0.0659 |
| | | 242 | 0.9797 | 0.9840 | 0.0581 |
| | MC dropout | 42 | 0.9534 | 0.9663 | 0.1248 |
| | | 142 | 0.9520 | 0.9619 | 0.1322 |
| | | 242 | 0.9574 | 0.9677 | 0.1063 |
| | QUAM | 42 | 0.9920 | 0.9930 | 0.0274 |
| | | 142 | 0.9900 | 0.9909 | 0.0348 |
| | | 242 | 0.9906 | 0.9915 | 0.0306 |

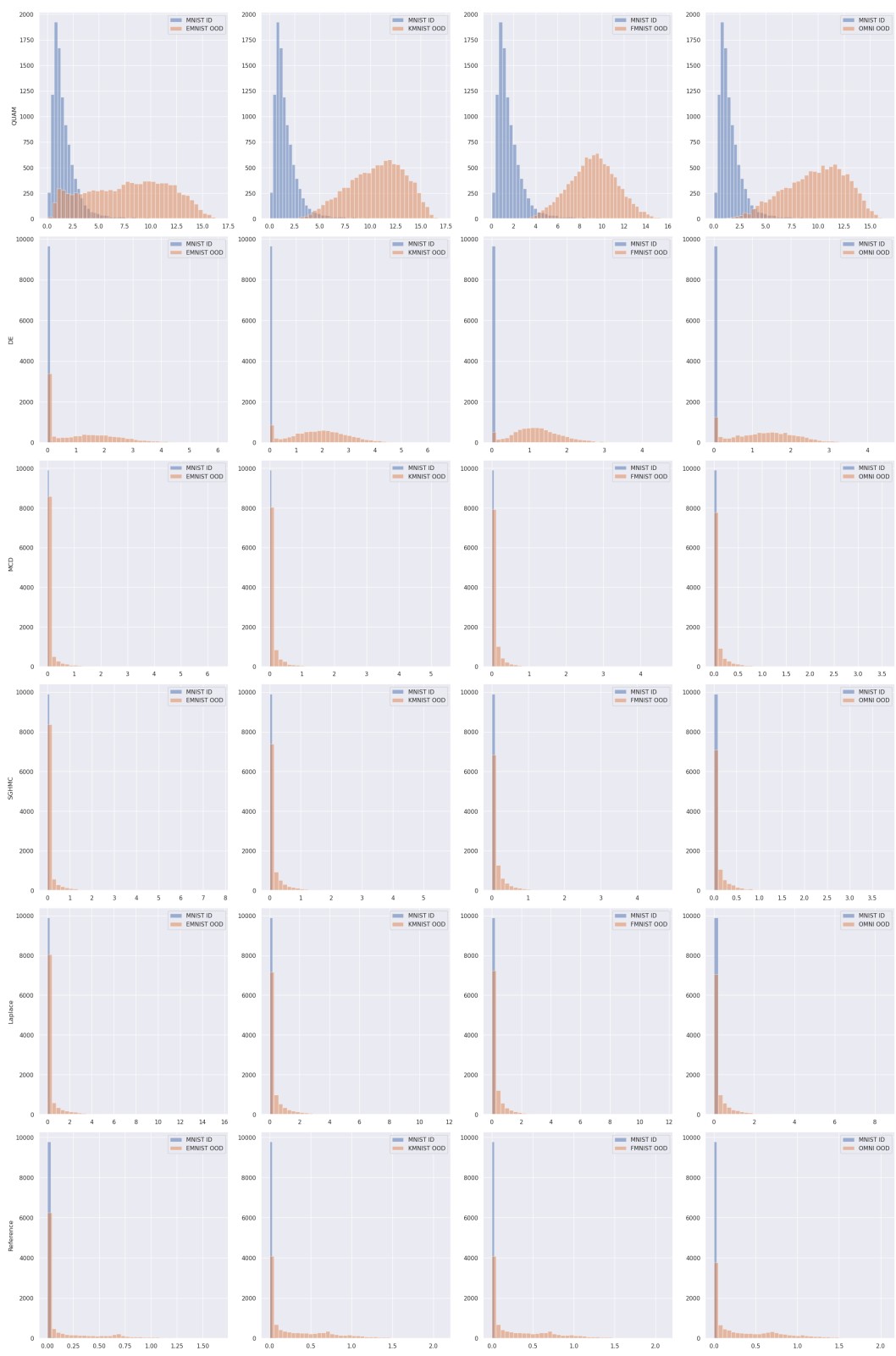

Figure C.5: MNIST: Histograms of uncertainty scores calculated for test set samples of the specified datasets.

Table C.3: Calibration: expected calibration error (ECE) based on the weighted average predictive distribution. Reference refers to the predictive distribution of the given, pre-selected model. Experiment was performed on three distinct splits, each containing 7000 ImageNet-1K validation samples.

| Reference | cSG-HMC | MCD | DE | QUAM |
|-----------|---------|-----|-----|------|
| $.159_{\pm.004}$ | $.364_{\pm.001}$ | $.166_{\pm.004}$ | $.194_{\pm.004}$ | $\mathbf{.096}_{\pm.006}$ |

(a) Reference    (b) cSG-HMC    (c) MCD    (d) DE    (e) QUAM

Figure C.6: Calibration: confidence vs. accuracy based on (weighted) average predictive distribution of different uncertainty quantification methods. Point size indicates number of samples in the bin.

against MC dropout used with 2048 samples with a dropout probability of 20%. The EfficientNet architectures utilize dropout only before the last layer. The adversarial model search for QUAM was performed on the last layer of the EfficientNet, which has 1.3 million trainable parameters. To enhance the computational efficiency, the output of the second-to-last layer was computed once for all samples, and this output was subsequently used as input for the final layer when performing the adversarial model search. We fixed $c_0$ to 1 and exponentially updated it at every of the 256 update steps. Also, weight decay was fixed to $1.e\text{-}4$ for the Adam optimizer [Kingma and Ba, 2014].

Two hyperparameters have jointly been optimized on ImageNet-O and ImageNet-A using a small grid search, with learning rate $\alpha \in \{5.e\text{-}3, 1.e\text{-}3, 5.e\text{-}4, 1.e\text{-}4\}$ and the exponential schedule update constant $\eta \in \{1.15, 1.01, 1.005, 1.001\}$. The hyperparameters $\alpha = 1.e\text{-}3$ and $\eta = 1.01$ resulted in the overall highest performance and have thus jointly been used for each of the three experiments. This implies that $c_0$ increases by 1% after each update step. We additionally searched for the best temperature and the best number of update steps for each experiment separately. The best temperature for scaling the cross-entropy loss when calculating Eq. (4) was identified as 0.05, 0.005, and 0.0005, while the best number of update steps was identified as 50, 100, and 100 for ImageNet-O OOD detection, ImageNet-A adversarial example detection, and ImageNet1K misclassification detection, respectively. Selective prediction was performed using the same hyperparameters as misclassification detection. We observed that the adversarial model search is relatively stable with respect to these hyperparameters.

The detailed results on various metrics and replicates of the experiments can be found in C.4. Histograms of the scores on the ID dataset and the OOD dataset, the adversarial example dataset and the correctly and incorrectly classified samples are depicted in Fig. C.7 for all methods. ROC curves, as well as accuracy over retained sample curves, are depicted in Fig. C.8. To provide confidence intervals, we performed all experiments on three distinct dataset splits of the ID datasets, matching the number of OOD samples. Therefore we used three times 2000 ID samples for Imagenet-O and three times 7000 ID samples for Imagenet-A and misclassification detection as well as selective prediction.

**Calibration.** Additionally, we analyze the calibration of QUAM compared to other baseline methods. Therefore, we compute the expected calibration error (ECE) [Guo et al., 2017] on the ImageNet-1K validation dataset using the expected predictive distribution. Regarding QUAM, the predictive distribution was optained using the same hyperparameters as for misclassification detection reported above. We find that QUAM improves upon the other considered baseline methods, although it was not directly designed to improve the calibration of the predictive distribution. Tab. C.3 states the ECE of considered uncertainty quantification methods and in Fig. C.6 the accuracy and number of samples (depicted by the size) for specific confidence bins is depicted.

Table C.4: Detailed results of ImageNet OOD detection, adversarial example detection and misclassification experiments, reporting AUROC, AUPR and FPR@TPR=95% for individual splits.

| OOD dataset / task | Method | Split | ↑ AUPR | ↑ AUROC | ↓ FPR@TPR=95% |
|---|---|---|---|---|---|
| ImageNet-O | Reference | I | 0.615 | 0.629 | 0.952 |
| | | II | 0.600 | 0.622 | 0.953 |
| | | III | 0.613 | 0.628 | 0.954 |
| | cSG-HMC | I | 0.671 | 0.682 | 0.855 |
| | | II | 0.661 | 0.671 | 0.876 |
| | | III | 0.674 | 0.679 | 0.872 |
| | MC dropout | I | 0.684 | 0.681 | 0.975 |
| | | II | 0.675 | 0.677 | 0.974 |
| | | III | 0.689 | 0.681 | 0.972 |
| | Deep Ensembles (LL) | I | 0.573 | 0.557 | 0.920 |
| | | II | 0.566 | 0.562 | 0.916 |
| | | III | 0.573 | 0.566 | 0.928 |
| | Deep Ensembles (all) | I | 0.679 | 0.713 | 0.779 |
| | | II | 0.667 | 0.703 | 0.787 |
| | | III | 0.674 | 0.710 | 0.786 |
| | QUAM | I | 0.729 | 0.758 | 0.766 |
| | | II | 0.713 | 0.740 | 0.786 |
| | | III | 0.734 | 0.761 | 0.764 |
| ImageNet-A | Reference | I | 0.779 | 0.795 | 0.837 |
| | | II | 0.774 | 0.791 | 0.838 |
| | | III | 0.771 | 0.790 | 0.844 |
| | cSG-HMC | I | 0.800 | 0.800 | 0.785 |
| | | II | 0.803 | 0.800 | 0.785 |
| | | III | 0.799 | 0.798 | 0.783 |
| | MC dropout | I | 0.835 | 0.828 | 0.748 |
| | | II | 0.832 | 0.828 | 0.740 |
| | | III | 0.826 | 0.825 | 0.740 |
| | Deep Ensembles (LL) | I | 0.724 | 0.687 | 0.844 |
| | | II | 0.723 | 0.685 | 0.840 |
| | | IIII | 0.721 | 0.686 | 0.838 |
| | Deep Ensembles (all) | I | 0.824 | 0.870 | 0.385 |
| | | II | 0.837 | 0.877 | 0.374 |
| | | III | 0.832 | 0.875 | 0.375 |
| | QUAM | I | 0.859 | 0.875 | 0.470 |
| | | II | 0.856 | 0.872 | 0.466 |
| | | III | 0.850 | 0.870 | 0.461 |
| Misclassification | Reference | I | 0.623 | 0.863 | 0.590 |
| | | II | 0.627 | 0.875 | 0.554 |
| | | III | 0.628 | 0.864 | 0.595 |
| | cSG-HMC | I | 0.478 | 0.779 | 0.755 |
| | | II | 0.483 | 0.779 | 0.752 |
| | | III | 0.458 | 0.759 | 0.780 |
| | MC dropout | I | 0.514 | 0.788 | 0.719 |
| | | II | 0.500 | 0.812 | 0.704 |
| | | III | 0.491 | 0.788 | 0.703 |
| | Deep Ensembles (LL) | I | 0.452 | 0.665 | 0.824 |
| | | II | 0.421 | 0.657 | 0.816 |
| | | III | 0.425 | 0.647 | 0.815 |
| | Deep Ensembles (all) | I | 0.282 | 0.770 | 0.663 |
| | | II | 0.308 | 0.784 | 0.650 |
| | | III | 0.310 | 0.786 | 0.617 |
| | QUAM | I | 0.644 | 0.901 | 0.451 |
| | | II | 0.668 | 0.914 | 0.305 |
| | | III | 0.639 | 0.898 | 0.399 |

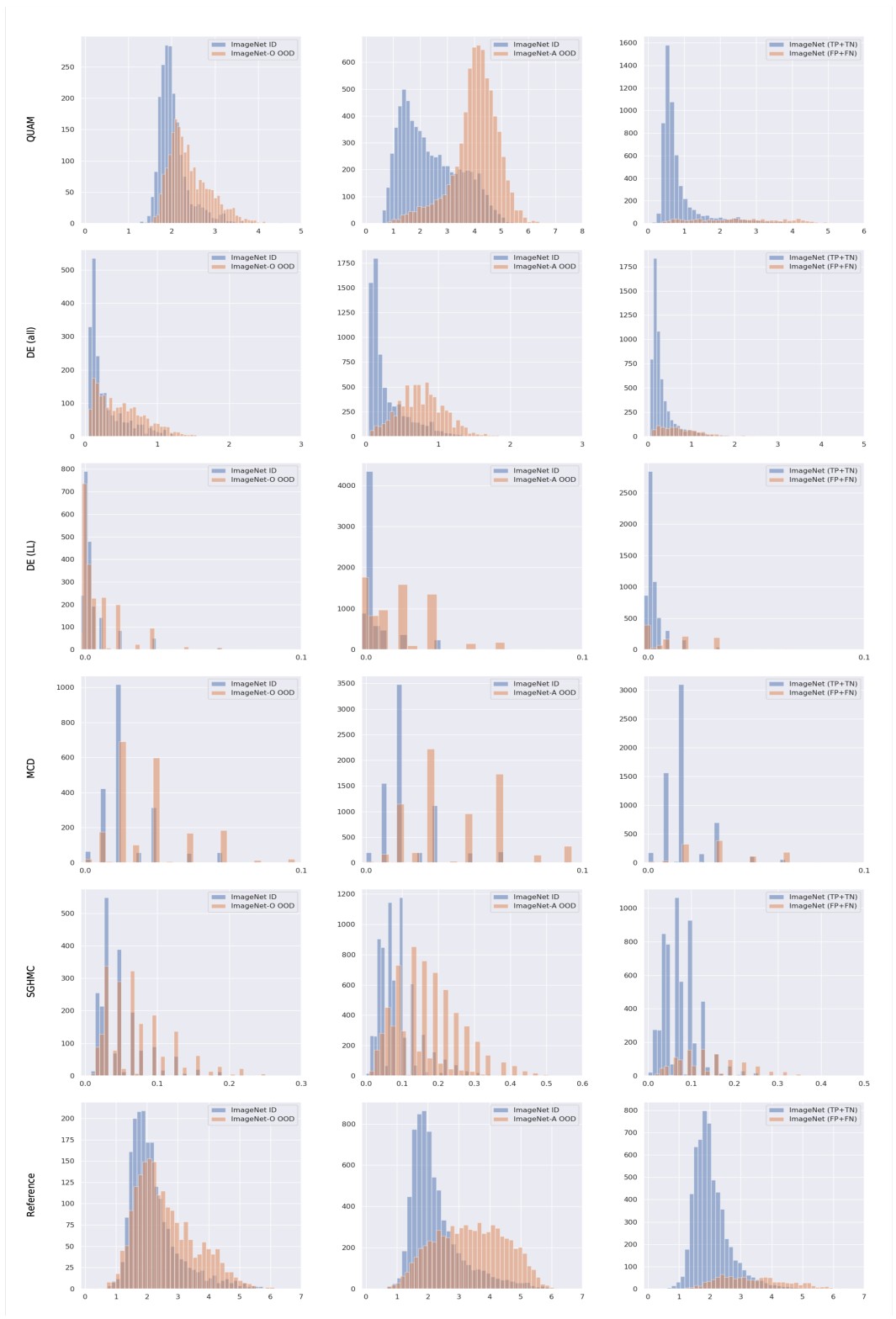

Figure C.7: ImageNet: Histograms of uncertainty scores calculated for test set samples of the specified datasets.

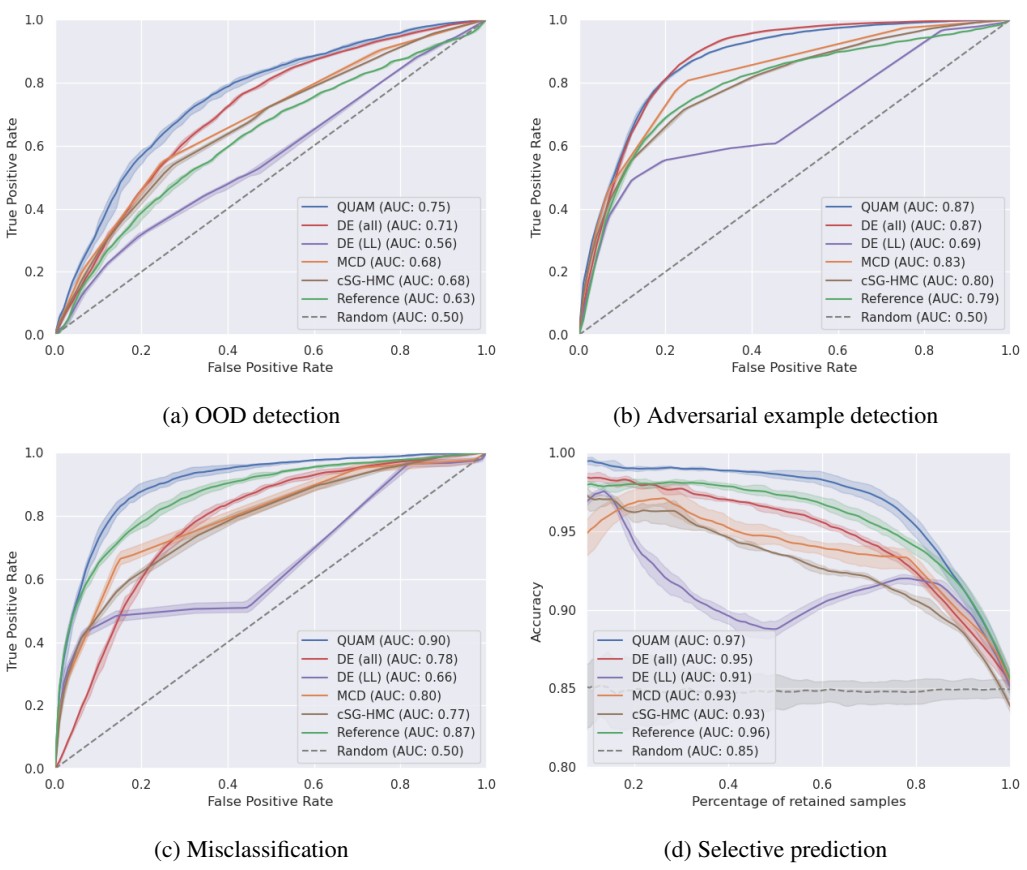

(a) OOD detection

(b) Adversarial example detection

(c) Misclassification

(d) Selective prediction

Figure C.8: ImageNet-1K OOD detection results on ImageNet-O, adversarial example detection results on ImageNet-A, misclassification detection and selective prediction results on the validation dataset. ROC curves using the epistemic uncertainty of a given, pre-selected model (as in Eq. (2)) to distinguish between (a) the ImageNet-1K validation dataset and ImageNet-O, (b) the ImageNet-1K validation dataset and ImageNet-A and (c) the reference model's correctly and incorrectly classified samples. (d) Accuracy of reference model on subset composed of samples that exhibit lowest epistemic uncertainty.

## C.5 Comparing Mechanistic Similarity of Deep Ensembles vs. Adversarial Models

The experiments were performed on MNIST, EMNIST, and KMNIST test datasets, using 512 images of each using Deep Ensembles, and the reference model $w$, trained on MNIST. Results are depicted in Fig. C.9. For each image and each ensemble member, gradients were integrated over 64 steps from 64 different random normal sampled baselines for extra robustness [Sundararajan et al., 2017]. Since the procedure was also performed on the OOD sets as well as our general focus on uncertainty estimation, no true labels were used for the gradient computation. Instead, predictions of ensemble members for which the attributions were computed were used as targets. Principal Component Analysis (PCA) was performed for the attributions of each image separately, where for each pixel the attributions from different ensemble members were treated as features. The ratios of explained variance, which are normalized to sum up to one, are collected from each component. If all ensemble members would utilize mutually exclusive features for their prediction, all components would be weighted equally, leading to a straight line in the plots in the top row in Fig. C.9. Comparatively high values of the first principal component to the other components in the top row plots in Fig. C.9 indicate low diversity in features used by Deep Ensembles.

The procedure was performed similarly for an ensemble of adversarial models. The main difference was that for each image an ensemble produced as a result of an adversarial model search on that specific image was used. We observe, that ensembles of adversarial models utilize more dissimilar features, indicated by the decreased variance contribution of the first principal component. This is especially strong for ID data, but also noticeable for OOD data.

## C.6 Prediction Space Similarity of Deep Ensembles and Adversarial Models

In the following, ensembles members and adversarial models are analyzed in prediction space. We used the same Deep Ensembles as the one trained on MNIST for the OOD detection task described in Sec. C.4.1. Also, 10 adversarial models were retrieved from the reference model $w$ and a single OOD sample (KMNIST), following the same procedure as described in Sec. C.4.1.

For the analysis, PCA was applied to the flattened softmax output vectors of each of the 20 models applied to ID validation data. The resulting points represent the variance of the model's predictions across different principal components [Fort et al., 2019]. The results in Fig. C.10 show, that the convex hull of blue points representing adversarial models, in general, is much bigger than the convex hull of orange points representing ensemble members across the first four principal components, which explain 99.99% of the variance in prediction space. This implies that even though adversarial models achieve similar accuracy as Deep Ensembles on the validation set, they are capable of capturing more diversity in prediction space.

## C.7 Computational Expenses

**Experiments on Synthetic Datasets** The example in Sec. C.2 was computed within half an hour on a GTX 1080 Ti. Experiments on synthetic datasets shown in Sec. C.3 were also performed on a single GTX 1080 Ti. Note that the HMC baseline took approximately 14 hours on 36 CPU cores for the classification task. All other methods except QUAM finish within minutes. QUAM scales with the number of test samples. Under the utilized parameters and 6400 test samples, QUAM computation took approximately 6 hours on a single GPU and under one hour for the regression task, where the number of test points is much smaller.

**Experiments on Vision Datasets** Computational Requirements for the vision domain experiments depend a lot on the exact utilization of the baseline methods. While Deep Ensembles can take a long time to train, depending on the ensemble size, we utilized either pre-trained networks for ensembling or only trained last layers, which significantly reduces the runtime. Noteworthy, MC-dropout can result in extremely high runtimes depending on the number of forward passes and depending on the realizable batch size for inputs. The same holds for SG-HMC. Executing the QUAM experiments on MNIST (Sec. C.4.1) took a grand total of around 120 GPU-hours on a variety of mostly older generation and low-power GPUs (P40, Titan V, T4), corresponding to roughly 4 GPU-seconds per sample. Executing the experiments on ImageNet (Sec. C.4.2) took about 100 GPU-hours on a mix of A100 and A40 GPUs, corresponding to around 45 GPU-seconds per sample. The experiments presented in Sec.C.5 and C.6 took around 2 hours each on 4 GTX 1080 Ti.

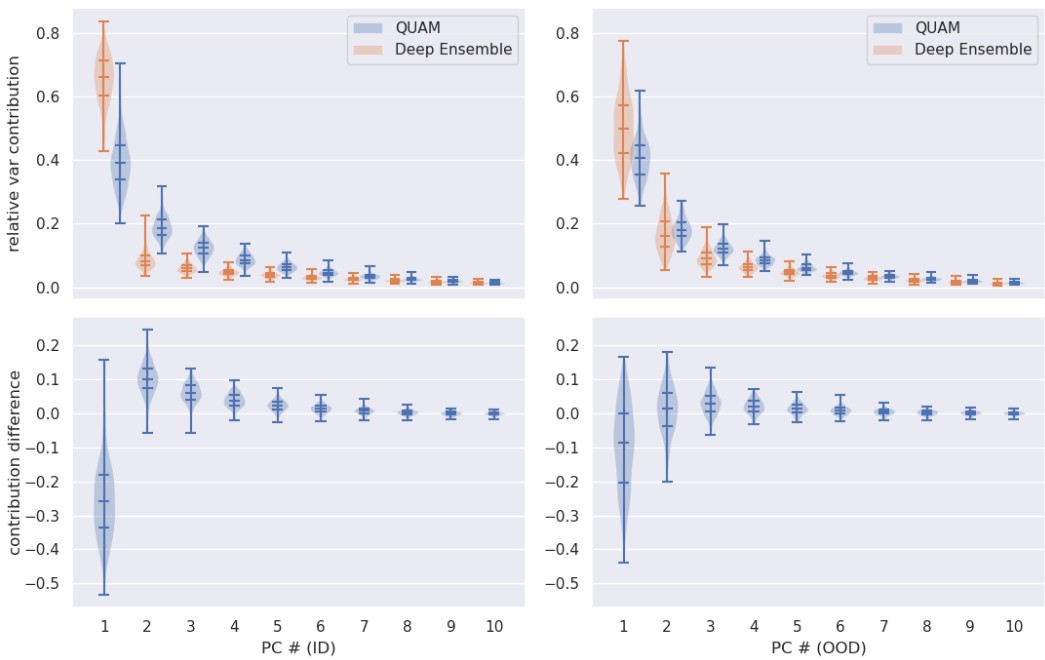

Figure C.9: The differences between significant component distribution are marginal on OOD data but pronounced on the ID data. The ID data would be subject to optimization by gradient descent during training, therefore the features are learned greedily and models are similar to each other mechanistically. We observe, that the members of Deep Ensembles show higher mechanistic similarity than the members of ensembles obtained from adversarial model search.

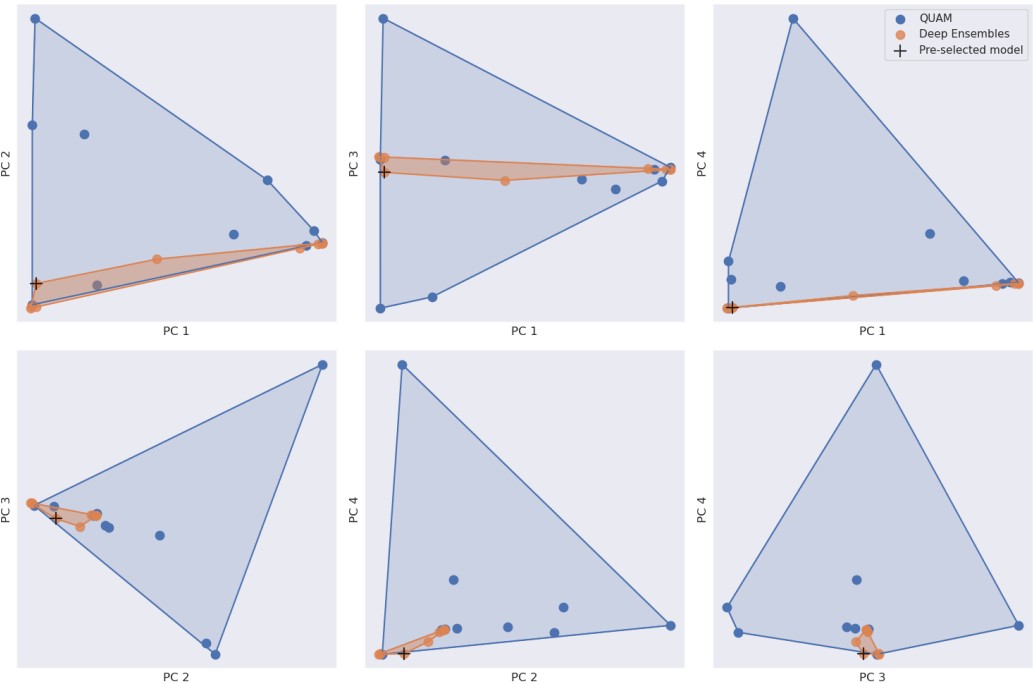

Figure C.10: Convex hull of the down-projected softmax output from 10 Ensemble Members (orange) as well as 10 adversarial models (blue). PCA is used for down-projection, all combinations of the first four principal components (99.99% variance explained) are plotted against each other. Softmax outputs are obtained on a batch of 10 random samples from the ID validation dataset. The black cross marks the given, pre-selected model $w$.

