# OpenReview forum: "Quantification of Uncertainty with Adversarial Models"
_NeurIPS.cc/2023/Conference — NeurIPS 2023 poster_

### Official Review · Reviewer_rwwy · 2023-07-01

**Soundness:** 3 good
**Presentation:** 3 good
**Contribution:** 2 fair
**Rating:** 4
**Confidence:** 3

**Summary:**

The authors propose a new method for estimating epistemic uncertainty. They propose to adversarially search for modes of the posterior distribution. Empirically, they demonstrate that their uncertainty performs well on OOD detection.

**Strengths:**

The empirical results of this approach on OOD detection are very promising.

**Weaknesses:**

- The method performs well on ood detection. Is the resulting uncertainty also well calibrated?
- l60 : The argument as to why ensembles and BNNs underperform is not clear to me. The authors say that they miss important posterior modes. Caen the authors elaborate on this?
- L142: can the authors demonstrate that the current methods underestimate uncertainty? Just because some methods dont explore all modes doesn mean that they underestimate uncertainty. E.g. they may overestimate the width of other modes.
- Lack of comparison with other ensemble methods, e.g. [1]. [1] in particular shows similar results on the two moon example. This makes me wonder whether the main impact of the method comes from the mixture of gaussians, as done in [1], or from the adversarial search. I guess it would be important to quantify that.

[1] Mixtures of Laplace Approximations for Improved Post-Hoc Uncertainty in Deep Learning


**Questions:**

see weaknesses

---

> ### Author Rebuttal · Authors · 2023-08-08
>
> We thank the reviewer for this assessment of our work. Regarding the stated weaknesses:
> - **Calibration:** Thank you for proposing this interesting direction. Your intuition was right, we found that our method indeed improves upon the other considered baseline methods, although it was not directly designed to improve calibration of the predictive distribution. The detailed results measuring the expected calibration error (ECE) and the calibration plots for the ImageNet-1K validation dataset are given in Table 2 and Figure 2 of the PDF document attached to the global answer. The results for QUAM were obtained using the same hyperparameters as for misclassification detection reported in the appendix of the main paper. Future work should explore methods based upon QUAM that specifically target to improve the calibration.
> - **Why do prior methods underestimate epistemic uncertainty:** In line 60 we raise the concern that Deep Ensembles and MC dropout fail to capture important posterior modes and thus underestimate epistemic uncertainty. This is inspired by [1], where the authors raise attention to the fact that “Ultimately, the goal is to accurately compute the predictive distribution in Eq. (1), rather than find a generally accurate representation of the posterior. In particular, we must carefully represent the posterior in regions that will make the greatest contributions to the BMA integral.”. The regions that contribute most to the posterior integrals defining the epistemic uncertainty in equations (1) and (2) in our paper are those where predictions differ a lot to the reference model, thus exhibit high KL-divergence. QUAM explicitly searches for those, while Deep Ensembles and MC dropout do not have an explicit mechanism to enforce variety in the predictive distributions on a new test sample, resulting in low values of KL-divergences. An additional reference in the literature would be [2], which also discusses the problem of functionally similar ensemble members for estimating the BMA predictive distribution. As pointed out also in [3], the issue is that Deep Ensembles maximize the posterior probability of the ensemble models through gradient descent, which is prone to yield functionally similar “easy” solutions. We are thus not concerned that prior methods do not find all modes, but modes that yield functionally different solutions that predict differently, thus underestimate the epistemic uncertainty. The same arguments apply also to the dropout models used in MC dropout.
> - **Empirical evidence that prior methods underestimate epistemic uncertainty:** This issue is empirically exemplified for Deep Ensembles and MC dropout on a toy dataset, where results are shown in Figure 2 in the main paper. They underestimate the epistemic uncertainty due to their lack of finding functionally different solutions that yield different predictions, compared to the ground truth (HMC) shown in Figure C.2 in the appendix. Furthermore, we observe the same issue in the experiments on the two-moon dataset (Figure 3) when considering a test point that is for instance in the upper left or lower right corner, compared to the ground truth (HMC). Here we directly observe that the epistemic uncertainty is too low. Similarly, we find that epistemic uncertainty is underestimated for inputs outside the range of data points (Figure C.4 in the appendix) in a regression task.
> - **Lack of comparison to other ensemble methods:** Thank you for pointing us to this interesting paper [4]. Unfortunately, the authors did not provide the original code for their proposed method MoLA. We gave our best effort to reproduce their results on the two-moon example using the reported pipeline, but found the results to only marginally differ from the single network Laplace approximation reported in our paper that uses the same implementation for the Laplace approximation as [4]. However, we also evaluated our reimplementation of MoLA on the MNIST OOD detection task. The results are stated in Table 3 in the attached pdf to our global answer. We found that MoLA improves upon the Laplace approximation of a single model, but leads to worse performance than just using the underlying Deep Ensemble and consequently does not outperform QUAM as well. We will release an updated version of our code including this new baseline and add the additional results to Table 1 for the final version of the paper.
>
> Thank you once again for your thoughtful assessment and valuable feedback. We have worked diligently to address your concerns, and we hope that our responses clarify the issues you raised. Please do not hesitate to reach out if you have any further questions or require additional information.
>
> ---
>
> [1] Wilson, A. G., & Izmailov, P. (2020). Bayesian deep learning and a probabilistic perspective of generalization. Advances in neural information processing systems, 33, 4697-4708.
>
> [2] D'Angelo, F., & Fortuin, V. (2021). Repulsive deep ensembles are bayesian. Advances in Neural Information Processing Systems, 34, 3451-3465.
>
> [3] Parker-Holder, J., Metz, L., Resnick, C., Hu, H., Lerer, A., Letcher, A., ... & Foerster, J. (2020). Ridge rider: Finding diverse solutions by following eigenvectors of the hessian. Advances in Neural Information Processing Systems, 33, 753-765.
>
> [4] Eschenhagen, R., Daxberger, E., Hennig, P., & Kristiadi, A. (2021). Mixtures of Laplace approximations for improved post-hoc uncertainty in deep learning. arXiv preprint arXiv:2111.03577.

---

> > ### Comment · Reviewer_rwwy · 2023-08-14
> >
> > Thank you or addressing my points. Further comments below:
> >
> > **Calibration**
> >
> > The plot 2 e) in the additional PDF looks odd. Why does QUAM almost exclusively predict 100% confidence? How would the method compare on a dataset where calibration is less saturated?
> >
> > **Empirical evidence that prior methods underestimate epistemic uncertainty**
> >
> > I dont think that fig. 2/3 are sufficient empirical evidence that ensembles under estimate uncertainty. In particular, other works have found that Ensembles are better calibrated than MCD. It seems there is rather sth. of with the trained ensemble in this work. The fact that all ensembles members reach the same solution indicates that there is not suficient randomness in the process? Do the authors only vary the model initilization?

---

> > > ### Author Response · Authors · 2023-08-16
> > >
> > > We thank the reviewer for his additional comments and questions.
> > >
> > > **Calibration**
> > >
> > > We agree with the reviewer, that the results depicted in Figure 2 e) in the PDF attached to the global answer looks different to other considered methods. Our explanation of them is as follows. The calibration results for QUAM were obtained using the same hyperparameters as for misclassification detection. Naturally, this is a discriminative task, where low uncertainty (thus high confidence) should be assigned to the correctly classified samples and high uncertainty (thus low confidence) to the misclassified samples. QUAM excels at this, as stated in Table C.2 in the appendix (IN-Misclass.). Since the majority of samples are correctly classified, we expect that QUAM yields high confidence for those samples. Future work could shed more light on the nature of these observations. We note that, while related, quantifying the uncertainty defined by Equations (1) and (2) in our paper is not equivalent to attaining a well calibrated prediction. *We would like to emphasize that QUAM was not explicitly designed to improve calibration, but does so empirically.*
> > >
> > > We acknowledge that future work would need to investigate calibration of those methods for different datasets. However, we argue that ImageNet is a favorable dataset for calibration, as the accuracy of the classifier is at ~84% top-1 accuracy, thus also lower accuracy bins (to calculate the ECE) contain enough samples to get robust statistical estimates.
> > >
> > > **Empirical evidence that prior methods underestimate epistemic uncertainty**
> > >
> > > To the best of our knowledge, it is not clear cut whether MCD or Deep Ensembles are better calibrated (see e.g. Figure 2 test performance in [1]). The members of the Deep Ensemble in Figures 2 and 3 of our paper are indeed the same architecture trained from different random initializations, exactly as proposed in the original paper [2] and generally used as a baseline in e.g. [1], [3], [4] or [5]. Given that [4] (Figure 1) and [5] (Figure 1) report qualitatively the same results for Deep Ensembles as in Figure 3 of our paper, it suggests that the experiment was implemented correctly. Also, equivalent results for the same experiment with Deep Ensembles have been reported in [3] (Figure 1), which was suggested by the reviewer. Further empirical evidence that MCD and ensembles of MCD models (a combination of Deep Ensembles and MCD thereof) underestimate epistemic uncertainty for regions far from training samples (in the latent space of a variational autoencoder trained on MNIST) is given in Figures 1 and 6 in [6]. Finally, note that we varied the model architecture by using different sizes of externally obtained, pre-trained, and verified EfficientNets [7] to perform the ImageNet experiments, as reported in Table 2 of our paper. QUAM empirically outperforms this ensemble in estimating the uncertainty as well.
> > >
> > > We hope that our response clarifies the remaining questions and gives rise to a more positive assessment of our work.
> > >
> > > ---
> > >
> > > [1] Ovadia, Y., Fertig, E., Ren, J., Nado, Z., Sculley, D., Nowozin, S., ... & Snoek, J. (2019). Can you trust your model's uncertainty? evaluating predictive uncertainty under dataset shift. Advances in neural information processing systems, 32.
> > >
> > > [2] Lakshminarayanan, B., Pritzel, A., & Blundell, C. (2017). Simple and scalable predictive uncertainty estimation using deep ensembles. Advances in neural information processing systems, 30.
> > >
> > > [3] Eschenhagen, R., Daxberger, E., Hennig, P., & Kristiadi, A. (2021). Mixtures of Laplace approximations for improved post-hoc uncertainty in deep learning. arXiv preprint arXiv:2111.03577.
> > >
> > > [4] Liu, J., Lin, Z., Padhy, S., Tran, D., Bedrax Weiss, T., & Lakshminarayanan, B. (2020). Simple and principled uncertainty estimation with deterministic deep learning via distance awareness. Advances in Neural Information Processing Systems, 33, 7498-7512.
> > >
> > > [5] Van Amersfoort, J., Smith, L., Teh, Y. W., & Gal, Y. (2020, November). Uncertainty estimation using a single deep deterministic neural network. In International conference on machine learning (pp. 9690-9700). PMLR.
> > >
> > > [6] Smith, L., & Gal, Y. (2018). Understanding measures of uncertainty for adversarial example detection. The Conference on Uncertainty in Artificial Intelligence (UAI).
> > >
> > > [7] Tan, M., & Le, Q. (2019, May). Efficientnet: Rethinking model scaling for convolutional neural networks. In International conference on machine learning (pp. 6105-6114). PMLR.

---

### Official Review · Reviewer_DS26 · 2023-07-01

**Soundness:** 4 excellent
**Presentation:** 4 excellent
**Contribution:** 4 excellent
**Rating:** 8
**Confidence:** 3

**Summary:**

The paper introduces Quantification of Uncertainty with Adversarial Models (QUAM), a novel approach for epistemic uncertainty estimation in deep learning. Well-known uncertainty quantification approaches, such as Deep Ensembles or variational inference, underestimate the epistemic uncertainty by sampling from posterior modes found by gradient descent, which might prevent some modes to be found due to the shape of the loss landscape. To overcome this limitation, QUAM introduces adversarial models: plausible models with a high posterior that differ in the prediction from a reference one. Mixture importance sampling, with the adversarial models as modes, is then used to estimate the epistemic uncertainty. Experiments on both synthetic and vision data confirmed the validity of the approach in improving epistemic uncertainty estimation.

**Strengths:**

The paper is very well-written, all the design choices are motivated by references to existing work, and the experiments are reproducible and robust. The topic is of great significance since in relates to important topics such as Out-Of-Distribution detection and Responsible AI, as also stated by the authors in the Societal Impact Statement. Overall, I really liked reading this work and I rate it as a strong accept.

**Weaknesses:**

I have no major remarks against this version of the paper.

**Questions:**

I have just two minor observations:
- The notation used in equation 1 (like the mutual information) was introduced only in the Appendix.
- Figure C.1 (Appendix): maybe wrong colour assignment of the adversarial model (yellow instead of blue).

**Limitations:**

The authors addressed potential negative societal impact of their work in the Appendix.

---

> ### Author Rebuttal · Authors · 2023-08-07
>
> We thank the reviewer for this very positive assessment of our work.
>
> Indeed we missed out on formally introducing the symbols for the cross-entropy, the KL-divergence and the mutual information in equation (1) in the main paper. Thank you for pointing out we will correct this in the final version of the paper.
>
> Furthermore, thank you for pointing out the wrong color assignment in the caption of Figure C.1 in the appendix. We changed the color of the adversarial model part from yellow to blue for better readability when preparing the final version of the supplementary material, but did not change the caption accordingly. We will correct that in the final version of the paper.

---

> > ### Comment · Reviewer_DS26 · 2023-08-17
> >
> > I thank the authors for having fixed my minor observations.
> >
> > Since all the reviewers acknowledged the contribution of adversarial models to the field, and the authors addressed most of the reviewers' concerns on clarity, missing related works and missing ablation studies, I confirm my positive assessment of the work.

---

### Official Review · Reviewer_EcJb · 2023-07-04

**Soundness:** 3 good
**Presentation:** 1 poor
**Contribution:** 3 good
**Rating:** 3
**Confidence:** 4

**Summary:**

This paper introduces Quantification of Uncertainty with Adversarial Models (QUAM). Building on the claim that previous epistemic uncertainty estimation methods (e.g. MC dropout, Deep Ensembles) underestimate the epistemic uncertainty by only considering the posterior distribution when sampling models, the authors introduce an Adversarial Model Search algorithm to identify models located within a posterior mode but with different predictive distributions compared to the reference model.

**Strengths:**

Previous epistemic uncertainty estimation methods (i) require changing the training procedure to account for uncertainty estimation by e.g. adding dropout layers or training multiple models, (ii) underestimate the epistemic uncertainty by only sampling models from the posterior distribution.

The proposed method QUAM addresses (i) by searching for adversarial models at test time for each test sample, thus working with any pre-trained model. Moreover, it solves (ii) by designing an adversarial model search algorithm that  allows to cover more posterior modes by identifying models located within a posterior mode but with different predictive distributions compared to the reference model.



**Weaknesses:**

## Lack of clarity
Although the method is interesting and addresses some important limitations of previous epistemic uncertainty estimation approaches, the paper writing should be revised.

Too often important details are omitted in the main paper or disclosed too late. In particular,
1. **Definition of adversarial models.** the definition of adversarial models introduced in this paper is completely new and has nothing to do with previous definitions of generative adversarial models [a, b] or adversarial attacks [c]. However, the first definition of what the authors mean by adversarial models is provided only at page 6, making it very hard to understand the paper before reading line 226. This paper would **greatly benefit** from earlier definitions of adversarial models in both abstract and introduction. Moreover, writing "(not adversarial examples!)" is the least elegant solution.
2. **Missing details.** Implementation details are solely reported in the supplement, but some details are essential in the main paper for better understanding. In particular, the number of adversarial models, number of MC samples, and the number of models in the deep ensemble should be clearly stated in the main paper.
3. **Settings (a) and (b)** It is not clear from the writing how the settings (a) and (b) defined in Section 2 relate to the experiments and whether the proposed QUAM presents any significant advantage on either task in particular compared to previous methods.

## Unsupported claims
Some claims are unsupported. They should either be supported by quantitative or theoretical analysis in the paper or some reference to other papers should be provided. In particular,
1. The claim that previous epistemic uncertainty estimation methods underestimate the uncertainty (line 59, line 143) is not backed up by empirical results or references to other papers.
2. "Adversarial models are characterized by a large value of the integrand of the integral that defines he epistemic uncertainty" (line 65). Without a definition of adversarial models in the introduction, this claim is not possible at all to understand.

## Missing ablation on number of adversarial models
The main paper does not specify the number of adversarial models learned through algorithm 1. According to the supplement, it seems that 10 are used. The paper would benefit from an ablation on the number of adversarial models

## Missing explanation of why the search algorithm does not always converge to the same solution
The search algorithm is not constrained to avoid finding always the same solution. Since it is an optimization problem, it is likely that similar solutions to the same problem are found if starting from the same initialization. The paper would benefit from an explanation of why the search does not converge always to the same adversarial model. Perhaps, enforcing that solutions must be different could improve the performance.

## Missing ablation on trade-off inference speed vs. uncertainty estimation performance at different number of samples
As pointed out by the authors, the proposed method sacrifices inference speed for uncertainty estimation performance. Although this undermines the practicality of the proposed approach, the theoretical insights gained from QUAM could outweigh this limitation.

However, an ablation on the trade-off between inference speed and uncertainty estimation performance at different number of samples for MC dropout, Deep Ensembles and QUAM is needed. I would expect MC dropout to be significantly more efficient for low number of samples, still obtaining reasonable performance at low inference time. Moreover, QUAM should have a higher uncertainty estimation performance upper bound than the competitors, if computational resources allow to tolerate the high inference time. This comparison would allow practitioners to understand which method to use based on their computation budget and needs.


## References
[a] Wang, Kunfeng, et al. "Generative adversarial networks: introduction and outlook." IEEE/CAA Journal of Automatica Sinica 4.4 (2017): 588-598.
[b] Li, Guofu, et al. "Security matters: A survey on adversarial machine learning." arXiv preprint arXiv:1810.07339 (2018).
[c] Chakraborty, Anirban, et al. "Adversarial attacks and defences: A survey." arXiv preprint arXiv:1810.00069 (2018).

**Questions:**

Overall, I find the paper unclear and lacking some important details and ablations. I believe the contribution to be relevant, but the paper would benefit from rewriting for clarity and from additional experiments based on the comments in the Weaknesses section.

I would change my opinion if the authors would convincingly address my comments in the Weaknesses section and, most importantly, optimize the paper for clarity. Providing the definition of adversarial models only at page 6 makes very hard understanding the first 5 pages.

**Limitations:**

The authors pointed out the limitations of their work.

---

> ### Author Rebuttal · Authors · 2023-08-08
>
> We thank the reviewer for this thoughtful feedback and critical assessment of our work, as well as the concrete suggestions to improve clarity. We will change the manuscript according to the reviewer’s suggestions for the final version as follows:
>
> - **Clarity:**
> 1) *Definition of adversarial models:* It is true that adversarial models are formally defined only at page 6, although they are a novel concept. In the abstract and the introduction, we only loosely define adversarial models as having both a high posterior as well as a high divergence between their predictions and that of a reference model, consequently being counterexamples of the reference model, predicting differently for a new input while explaining the training data equally well (Lines 12-13 and 65-70). This is because we deliberately have chosen to first focus on presenting the problem, only then followed by the solution, namely the search for adversarial models. However, to improve clarity, we will refine and expand these early definitions and add the formal definition to the introduction (after Line 70) in the final version of the paper.
> 2) *Missing details:* Thank you for pointing this out. We will move the number of adversarial models, number of MC samples, and the number of models in the Deep Ensemble from the appendix to the end of section 3 as a new subsection to further improve clarity.
> 3) *Settings (a) and (b):* QUAM is a method to estimate the integral defining the epistemic uncertainty in both setting (a) and (b), which are posterior expectations of divergences between predictive distributions. We evaluated the synthetic dataset experiments on both settings (a) and (b). Regarding the large scale experiments in the vision domain, we focused on our newly introduced setting (b). This is stated in the first sentence introducing this set of experiments (line 278/279) and in the table headers.
> - **Unsupported claims:**
> 1) Thank you for raising this point, we added references to similar claims in the literature as follows: [1] points out that adequate sampling of those posterior regions that make the greatest contribution to the posterior expectation of the predictive distribution are of greater interest for the purpose of estimating this posterior expectation than obtaining a generally good posterior representation. Those regions are the ones with the highest KL divergence in our setting (b) that only QUAM explicitly searches for. Similarly, [2] claims that Deep Ensembles do not provide enough functional diversity; we argue the same applies to MC dropout. [3] claims this is due to gradient descent always finding similar “easy” solutions.
> Empirically we investigated this issue on a toy dataset, results are shown in Figure 2, the ground truth (HMC) to compare to is shown in Figure C.2 in the appendix. Furthermore, we observe that epistemic uncertainty is underestimated in the experiments on the two-moon dataset (Figure 3) when considering a test point that is for instance in the upper left or lower right corner. Similarly, we find that epistemic uncertainty is underestimated for inputs outside the range of data points (Figure C.4 in the appendix) in a regression task.
> 2) We did not perceive this as a claim but solely as a loose version of the definition of adversarial models. Nonetheless, you are right that it is hard to understand at this stage in the paper. We will improve it by including the definition in the introduction as suggested before.
> - **Number of adversarial models and convergence:** We use 10 adversarial models for the MNIST experiments and 10 to 1000 for the ImageNet-1k experiments (see ablation study below). As you pointed out, directly maximizing the KL-divergence to the prediction of the reference model would indeed converge to very similar solutions, which we empirically observed during early experiments. Therefore, we instead minimize the cross-entropy towards one out of all possible classes at a time. This way we attain diverse solutions that, as you pointed out, lead to improved performance. This is discussed in detail in appendix section C.1. We will move these implementational details to the main paper at the end of section 3 as a new subsection to make it more explicit.
> - **Ablation on inference speed vs. performance:** QUAM requires updating the last layer for up to 100 steps for ImageNet-1k OOD detection, which is equivalent to 15 forward passes on the full network required by MC dropout (assuming that the last layer is 5\% of all parameters and that the backward pass is twice as expensive as the forward pass). In a new ablation study, we searched for adversarial models only on a subset of classes based on the highest softmax probabilities assigned by the given, pre-selected model. The results are listed in Table 1 and Figure 1 of the PDF document attached to the global answer. For instance, when searching adversarial models only for the 10 most probable classes (QUAM$\_{top1\\%}$), the number of full forward passes per sample is reduced from 15k (QUAM$\_{all}$) to 150, an inference speed reduction by a factor of 100. Still, QUAM outperforms MC dropout in terms of performance. Also, training a single additional ensemble member requires more computational cost than evaluating all ImageNet-1k OOD samples with QUAM. We will provide more details on inference speed vs. performance in the final version of the paper.
>
> We hope to have addressed your open questions and concerns. Your insightful review has guided us in enhancing the clarity of our paper, and we believe these changes will contribute positively to the overall impact of our work.
>
> ---
> [1] Wilson, A. G., & Izmailov, P. (2020). Bayesian deep learning and a probabilistic perspective of generalization. NeurIPS.
>
> [2] D'Angelo, F., & Fortuin, V. (2021). Repulsive deep ensembles are bayesian. NeurIPS.
>
> [3] Parker-Holder, J., & ... & Foerster, J. (2020). Ridge rider: Finding diverse solutions by following eigenvectors of the hessian. NeurIPS.

---

> > ### Comment · Reviewer_EcJb · 2023-08-13
> >
> > The authors provided a convincing rebuttal, and I appreciate their efforts towards improved clarity of their submission.
> >
> > - **Clarity:**
> > 1. _Definition of adversarial models:_ I thank the authors for acknowledging the importance of an earlier definition of adversarial models, and for adding it to the introduction.
> > 2. _Missing details:_ This is an important step for clarity and reproducibility.
> >
> > - **Unsupported Claims:**
> > The reply provided by the authors and supporting references seem convincing.
> >
> >
> > - **Number of adversarial models and convergence:**
> > I thank the authors for pointing out Sec. C. 1 in the appendix. Sec. C.1 reports fundamental details that should at least partially be mentioned in the main paper. The observation that "directly maximizing the KL divergence always leads to similar solutions to the optimization problem" was an obvious limitation by just reading the main paper, and it was not addressed there. To ensure that this paper complies with the clarity standards of this conference, I hope that the authors will carefully discuss this in the main paper.
> >
> > - **Ablation on inference speed vs. performance:** The reply provided by the authors, the new proposal for tuning the computation cost of QUAM and the results provided in Figure 1 are convincing. I thank the authors for providing them and I believe that they will enrich the paper.
> >
> > Overall, the authors addressed most of my concerns. However, my initially negative opinion was also greatly influenced by the initial poor quality of presentation and clarity. The authors will have to work on a significant rewriting to make the paper suitable for the standards of this conference.
> >
> > I am leaning towards increasing my rating, the extent of which will also depend on the discussion with other reviewers.

---

> > > ### Author Response · Authors · 2023-08-16
> > >
> > > Thank you again for your helpful and constructive suggestions. We rewrote the manuscript accordingly, which resulted in a significant improvement of clarity and quality of presentation. We fully agree with the reviewer that the clarity and good presentation of research results are very important. To summarize, we did the following main improvements of the manuscript:
> > > - **Number of adversarial models and convergence:** We added a new subsection on the practical implementation at the end of section 3 in the main paper. This subsection is based upon section C.1 from the appendix. Most importantly, we now discuss the problem that direct KL optimization always leads to the same solution as explained in Sec. C.1. We further explain that we want to find different regions with large contributions to the epistemic uncertainty integral in equation (1) and (2). This is achieved by minimizing the cross-entropy towards one out of all possible classes at a time. Indeed, the more of those regions QUAM identifies, the more effective is mixture importance sampling. We agree with the reviewer that it is essential to elaborate on this problem and how we tackled it in the main paper, as it is essential to our method.
> > > - **Definition of adversarial models:** We added Definition 1 to the introduction (paragraph starting at line 58). Consequently, we revised the informal definition of adversarial models previously given in this paragraph accordingly. Furthermore, we contrast the definition of adversarial models from other concepts with ‘adversarial’ in their naming, such as “adversarial examples”, “adversarial training”, “generative adversarial networks”, or “adversarial model-based RL”.
> > > - **Unsupported claims:** We added the additional references [1], [2] and [3] and extended discussion of why prior methods underestimate epistemic uncertainty to the respective claims (line 59, line 143) and more explicitly refer to our empirical evidence.
> > > - **Missing details:** We moved information about the most crucial hyperparameters (e.g. ensemble size, # of forward passes for MC dropout, # adversarial model searches, …) from the appendix to the experimental section in the main paper.
> > > - **Ablation on inference speed vs. performance:** We added the new ablation on inference speed vs. performance to the main paper.
> > > - **Additional experiments requested by other reviewers:** We added the new results for MoLA to Table 1 in the main paper. Furthermore, we added the results for calibration to the appendix.
> > >
> > > Would the reviewer see the need for any additional significant changes? We would gladly consider and address those.
> > >
> > > Finally, we thank the reviewer for acknowledging our convincing results, as well as our efforts towards improving the clarity and quality of presentation of our paper and hope that they adjust their assessment of our work accordingly.
> > >
> > > ---
> > >
> > > [1] Wilson, A. G., & Izmailov, P. (2020). Bayesian deep learning and a probabilistic perspective of generalization. NeurIPS.
> > >
> > > [2] D'Angelo, F., & Fortuin, V. (2021). Repulsive deep ensembles are bayesian. NeurIPS.
> > >
> > > [3] Parker-Holder, J., & ... & Foerster, J. (2020). Ridge rider: Finding diverse solutions by following eigenvectors of the hessian. NeurIPS.

---

### Official Review · Reviewer_T3Ph · 2023-07-07

**Soundness:** 4 excellent
**Presentation:** 4 excellent
**Contribution:** 4 excellent
**Rating:** 7
**Confidence:** 5

**Summary:**


This paper is about uncertainty estimation using adversarial models (not examples!).  The authors propose a new uncertainty estimation method, called QUAM, which performs a search of an adversarial model, which is one that fits the training set but has predictions far away from the predefined model, with the idea that a point is uncertain if multiple models explain it very differently, while still fitting the original training set.

QUAM has the potential to estimate epistemic uncertainty better than other uncertainty models, and experimental results point in this direction.

Contributions are:
- The QUAM framework for uncertainty quantification using adversarial model search.
- The new concept of an adversarial model, different from an adversarial example, which is a model (set of weights) that explains the training set well while having maximum difference with the orignal model in a new test point.
- The proposed method can estimate epistemic uncertainty for any model, including a pretrained model, which is an advantage over usual uncertainty methods that require modifications to the training process or model retraining.

**Strengths:**

- The paper is very well written and easy to understand.
- The paper touches an important topic, uncertainty estimation methods fail in samples far from the training set, usually producing overconfident uncertainties that are not useful to detect out of distribution samples. This is well known from Ovadia et al and other papers in the literature.
- This paper defines a new kind of meta-model, the adversarial model for uncertainty quantification, where to make a prediction in a new test point, new model parameters are found that fit the training set while providing alternative explanations for the new test point, producing higher quality epistemic uncertainty. This is of course very computationally expensive and it is mentioned as a limitation in the conclusions.
- Experimental results show that QUAM outperforms the baselines on the task of out of distribution detection on MNIST vs FMNIST/KMINST/EMNIST/OMNIGLIT and  ImageNet-1K vs Imagenet-O/ImageNet-A.
- I believe the proposed method QUAM is a good contribution to the field of uncertainty quantification, as it proposed a new framework for uncertainty estimation that is model agnostic and provides high quality epistemic uncertainty estimates.

**Weaknesses:**

- I have some serious doubts about the concept/interpretation of aleatoric uncertainty in this paper. In the literature and widely agreed concepts, aleatoric uncertainty is about the data, it is a property of the data, like stochasticity in measuring processes or noise, but the paper makes claims that "aleatoric uncertainty is the stochasticity of the model and epistemic uncertainty is the uncertainty about model parameters." (Lines 39/40), here the  claim is made that aleatoric uncertainty is about stochasticity of the model, which I do not believe it is correct. Just as an example, there are non-stochastic models (like ensembles) that can estimate aleatoric uncertainty without using stochastic components. And I mention previously, aleatoric uncertainty is mainly a property of the data, not of the model, even though a model can estimate the aleatoric uncertainty of the data. I believe here the paper should clarify or simply fix the definition.

~~- Another issue I have is about selection of baselines, in the two moons setting, DUQ [1] is a strong baseline that actually approximates epistemic uncertainty much closer than QUAM in the two moons dataset (see Figure 1 of the DUQ paper), and I believe this also points that an issue in epistemic uncertainty estimation is the model's inductive bias. DUQ uses a RBF layer to predict classes, based on distance to class centroids, unlike a separating hyperplane in a highly dimensional space, so epistemic uncertainty is very different. I believe in this paper DUQ and other single network uncertainty estimation method (like Direct Uncertainty Estimation, DDU), should also be compared.~~

**Questions:**

Questions:
- How were the baselines selected? The baseline UQ methods are not state of the art for epistemic uncertainty estimation. I already mentioned in weaknesses that DUQ has better epistemic uncertainty estimation in the two moons dataset.
- In Figure 3f, QUAM has artifacts or non-smoothness in the uncertainty, in the top-left and bottom-right corners, while other models do not have such artifacts, is there an explanation or intuition about this?

Post-rebutal, both of these questions have been answered to my satisfaction by the authors.

Suggestions:
- The DE (LL) setting is the same as the concept of sub-ensembles [2], maybe it could be mentioned or cited.
- In Tables 1 and 2, the aleatoric uncertainty of the model is claimed to be used, but I am not sure what this means, as most models actually output predictive uncertainty, which is the combination of aleatoric and epistemic uncertainty. Even a model without any uncertainty quantification (i.e without using MC-Dropout, Ensembling, BNNs, etc) still has a non-zero degree of aleatoric and epistemic uncertainty, so I believe this claim should be changed to predictive uncertainty and not aleatoric uncertainty.

[2]: Deep Sub-Ensembles for Fast Uncertainty Estimation in Image Classification
by Matias Valdenegro-Toro. arXiv 1910.08168.

**Limitations:**

The paper properly discusses the limitations of the proposed method in the conclusion sections, including the fact that for each new test point, a model search has to be performed which is computationally very expensive.

---

> ### Author Rebuttal · Authors · 2023-08-08
>
> We thank the reviewer for the thoughtful assessment of our work. Regarding the explicitly stated remarks and questions:
> - **Aleatoric uncertainty:** Your remark is correct, aleatoric uncertainty is indeed a property of the data, stemming from the stochasticity / noise in the measurement process as we point out in line 29/30. Fundamentally, aleatoric uncertainty is due to the stochasticity of the conditional distribution $p(\boldsymbol{y} \mid \boldsymbol{x})$ (often referred to as predictive distribution) that assigned labels to the samples in the dataset. However, we do not know the underlying conditional distribution, but use a model to approximate it, resulting in an approximation of the conditional distribution $p(\boldsymbol{y} \mid \boldsymbol{x}, \boldsymbol{w}) \approx p(\boldsymbol{y} \mid \boldsymbol{x})$. Therefore, we consider quantifying the aleatoric uncertainty as characterizing this stochasticity of the model, in accordance with e.g. [1] or [2]. Nevertheless, you are right that lines 39/40 should be improved. We suggest “Consequently, we consider uncertainty quantification as characterizing a stochastic model of the world. Here, aleatoric uncertainty stems from the stochasticity of the model and epistemic uncertainty from the variability of plausible model parameters.”, but would appreciate suggestions.
> - **Choice of baselines:** We certainly acknowledge the recent advances in single network uncertainty estimation like DDU [3], DUQ [4], DUE [5], SNGP [6], and others, and their ability to provide meaningful epistemic uncertainty estimates based on the location of new samples in the feature space. We also agree that they provide uncertainty estimates akin to those of HMC in the two-moon example. However, we did not compare our approach to these for the following reasons:
>   1) *Model Constraints:* The aforementioned single forward pass methods require the regularization of the feature space to assess uncertainty (either a two-sided gradient penalty or spectral normalization in the examples above), which has to be applied during training of the model. Therefore, it is not applicable under our newly introduced setting (b), where we assume a given, pre-selected model without any constraints of how it was obtained.
>   2) *Different Uncertainty Perspectives:* Single forward pass methods capture a different notion of epistemic uncertainty than Bayesian methods. They capture epistemic uncertainty through the location of new test samples in the latent feature space, whereas we consider capturing epistemic uncertainty through posterior integrals, thus through sampling different models. We solely claim that QUAM enhances the quantification of epistemic uncertainty for the specific notion given by the respective terms in equations (1) [setting (a)] and (2) [setting (b)] of the main paper. We thus selected the best known baselines that also aim to quantify this notion of uncertainty. A comprehensive exploration of the relation between these two different notions of epistemic uncertainty remains an exciting topic for future research, which hasn’t been deeply explored yet to the best of our knowledge.
> - **Artifacts / non-smoothness:** QUAM finds adversarial models for every input $\boldsymbol{x}$. Thus, to create Figure 3, we applied our method to each point of an input-mesh used to show the uncertainty over the whole input space. For all other methods, we applied the same sampled models to the whole input-mesh at once, thus the higher smoothness. Note that if e.g. MC dropout is applied to each test input individually instead of to all of them at once, the same non-smoothness will be observed due to the different sampled models for different test points.
>
> Thank you for pointing out the connection of the last layer ensemble to [7], we will reference it accordingly in the final version of the paper.
> Regarding the aleatoric uncertainty in Tables 1 and 2, we used the estimate of aleatoric uncertainty as in equation (2). This is the entropy of the predictive distribution under the single given, pre-selected model $\mathrm{H}[p(\boldsymbol{y} \mid \boldsymbol{x}, \boldsymbol{w})]$ and therefore solely an estimate of aleatoric uncertainty.
>
> We greatly appreciate your insightful feedback, which has helped us to refine and clarify our work. Should you have any further questions or need additional explanations, we look forward to provide them. Thank you for your time and consideration.
>
> ---
>
> [1] Helton, J. C. (1997). Uncertainty and sensitivity analysis in the presence of stochastic and subjective uncertainty. journal of statistical computation and simulation, 57(1-4), 3-76.
>
> [2] Kendall, A., & Gal, Y. (2017). What uncertainties do we need in bayesian deep learning for computer vision?. Advances in neural information processing systems, 30.
>
> [3] Mukhoti, J., Kirsch, A., van Amersfoort, J., Torr, P. H., & Gal, Y. (2021). Deep deterministic uncertainty: A simple baseline. arXiv preprint arXiv:2102.11582.
>
> [4] Van Amersfoort, J., Smith, L., Teh, Y. W., & Gal, Y. (2020, November). Uncertainty estimation using a single deep deterministic neural network. In International conference on machine learning (pp. 9690-9700). PMLR.
>
> [5] van Amersfoort, J., Smith, L., Jesson, A., Key, O., & Gal, Y. (2021). On feature collapse and deep kernel learning for single forward pass uncertainty. arXiv preprint arXiv:2102.11409.
>
> [6] Liu, J., Lin, Z., Padhy, S., Tran, D., Bedrax Weiss, T., & Lakshminarayanan, B. (2020). Simple and principled uncertainty estimation with deterministic deep learning via distance awareness. Advances in Neural Information Processing Systems, 33, 7498-7512.
>
> [7]: Valdenegro-Toro, M. (2019). Deep sub-ensembles for fast uncertainty estimation in image classification. arXiv preprint arXiv:1910.08168.

---

> > ### Comment · Reviewer_T3Ph · 2023-08-17
> > **good rebuttal**
> >
> > Thank you for the detailed rebuttal. I agree with point 2. About point 3, wouldn't it make more sense to then predict all baselines the same way, even if they have similar artifacts? At least to make a sensible comparison.
> >
> > About point 1, aleatoric uncertainty, I think here I have the biggest disagreement, I think your description is a bit conflicting because the conditional distribution has mixed aleatoric and epistemic uncertainty in it (called predictive uncertainty), and obtaining only aleatoric uncertainty from it is difficult without additional methods, that is why model stochasticity does not model aleatoric uncertainty exclusively. Since this point is not really relevant from your paper, I suggest to actually remove it since it could mislead future readers.
> >
> > I will update my review accordingly.

---

> > > ### Author Response · Authors · 2023-08-21
> > >
> > > We thank the reviewer for updating the review and the rating of our work.
> > >
> > > Regarding point 3, we agree with the reviewer that predicting all baselines the same way on a per-test-sample basis would make a better comparison to QUAM. However, previous work such as [1], [2] or [3] conducted similar experiments on the two-moon dataset, applying the baselines on all test samples at once. We decided to stay close to the setup of prior work, but we can also change the baselines to operate on a per-test-sample basis. Moreover, for a sufficiently high number of sampled models, the artifacts will smoothen out, so we could also just do more adversarial model searches for QUAM. Either way, we will elaborate on the origin of artifacts being different models sampled for individual test points.
> > >
> > > We fully agree with the reviewer, the debated point about aleatoric uncertainty is not really relevant for our paper and change the respective section accordingly.
> > >
> > > ---
> > >
> > > [1] Eschenhagen, R., Daxberger, E., Hennig, P., & Kristiadi, A. (2021). Mixtures of Laplace approximations for improved post-hoc uncertainty in deep learning. arXiv preprint arXiv:2111.03577.
> > >
> > > [2] Liu, J., Lin, Z., Padhy, S., Tran, D., Bedrax Weiss, T., & Lakshminarayanan, B. (2020). Simple and principled uncertainty estimation with deterministic deep learning via distance awareness. Advances in Neural Information Processing Systems, 33, 7498-7512.
> > >
> > > [3] Van Amersfoort, J., Smith, L., Teh, Y. W., & Gal, Y. (2020, November). Uncertainty estimation using a single deep deterministic neural network. In International conference on machine learning (pp. 9690-9700). PMLR.

---

### Author Rebuttal · Authors · 2023-08-08

We thank all reviewers again for the time and effort they have invested in order to provide their high quality feedback.


All reviewers found our approach novel and relevant, and acclaimed the general applicability of the method as well as the empirical performance. Nevertheless, reviewers were not unanimous about the paper's clarity of writing and requested additional ablations and evaluations, as well as further baselines for our OOD detection experiments.


We are confident to have addressed all concerns in the individual responses and will incorporate them in the final version of the paper. To summarize:

- We have supported the claim that prior methods underestimate uncertainty by citing several papers and further elaborated on the empirical evidence provided in our paper.
- We have improved the clarity of writing based on the reviewers feedback, e.g. by shifting implementational details from the appendix to the main paper and introducing the formal definition of adversarial models earlier.
- We have performed three additional experiments/ablations (see attached PDF document) that further confirm the strong performance of QUAM:
  1) Table 1 and Figure 1: We compared QUAM to other baselines in terms of inference speed (*as requested by reviewer EcJb*). We discovered that the inference time of QUAM can be reduced by a factor of 100, while outperforming baselines like MCD in both performance and speed.
  2) Table 2 and Figure 2: We evaluated QUAM in terms of expected calibration error (*as requested by reviewer rwwy*). Although QUAM was not directly designed to be well calibrated, we discovered that it outperforms our baseline methods in being able to provide a more calibrated output prediction.
  3) Table 3: We added a new baseline (MoLA) to our OOD detection experiment (*as requested by reviewer rwwy*). QUAM outperforms the proposed method in OOD detection on the FMNIST, KMNIST, EMNIST and OMNIGLOT test datasets.

We hope that this clarifies all questions and concerns and that the additional experiments provide convincing evidence of the effectiveness of our method. Thank you for your time and efforts!

---

### Comment · Area_Chair_hFiv · 2023-08-21
**Discussion with authors ends tomorrow**

Dear reviewers,

This is just a reminder that discussions with the authors will end tomorrow (Aug. 21).  Many of you have engaged active discussions with the authors.  If you have not, please read authors’ rebuttal and other reviews, and engage with the authors for any concerns you may still have.  Please remember to update your ratings after the discussions.

Best,
Your AC

---

### Decision · Program_Chairs · 2023-09-21

**Decision:**

Accept (poster)

**Comment:**

This paper receives 4 reviews,  including 1 strong accept, 1 accept, 1 reject, and 1 borderline reject, with an average rating of 5.5.  Reviewers find the idea of uncertainty estimation using adversarial search model novel, with improved performance over current epistemic uncertainty estimation methods.  Before rebuttal, they also identify some deficiencies with this work.  Their main concerns include unclear presentation, missing ablation study, missing important details, and lack of comparison with other ensemble methods.  The authors rebuttal was effective, largely addressing reviewers concerns.  Some minor concerns remain.  The authors are strongly advised to carefully revise the paper as per reviewer comments, in particular improving its presentation clarity and strengthening the experiments.